# HBO1 catalyzes lysine lactylation and mediates histone H3K9la to regulate gene transcription

Ziping Niu[1,6], Chen Chen [1,6] ✉, Siyu Wang[1,6], Congcong Lu[2,6], Zhiyue Wu[1], Aiyuan Wang[1], Jing Mo[3], Jianji Zhang[1], Yanpu Han[1], Ye Yuan[4], Yingao Zhang[1], Yong Zang[1], Chaoran He[1], Xue Bai[1], Shanshan Tian[1], Guijin Zhai[1], Xudong Wu [4] & Kai Zhang [1,5] ✉

Lysine lactylation (Kla) links metabolism and gene regulation and plays a key role in multiple biological processes. However, the regulatory mechanism and functional consequence of Kla remain to be explored. Here, we report that HBO1 functions as a lysine lactyltransferase to regulate transcription. We show that HBO1 catalyzes the addition of Kla in vitro and intracellularly, and E508 is a key site for the lactyltransferase activity of HBO1. Quantitative proteomic analysis further reveals 95 endogenous Kla sites targeted by HBO1, with the majority located on histones. Using site-specific antibodies, we find that HBO1 may preferentially catalyze histone H3K9la and scaffold proteins including JADE1 and BRPF2 can promote the enzymatic activity for histone Kla. Notably, CUT&Tag assays demonstrate that HBO1 is required for histone H3K9la on transcription start sites (TSSs). Besides, the regulated Kla can promote key signaling pathways and tumorigenesis, which is further supported by evaluating the malignant behaviors of HBO1- knockout (KO) tumor cells, as well as the level of histone H3K9la in clinical tissues. Our study reveals HBO1 serves as a lactyltransferase to mediate a histone Kla-dependent gene transcription.

As the end product of glucose metabolism, lactate has long been considered as a waste product under hypoxic conditions that causes harmful effects since its discovery in 1780s[1]. Recent study in 2019 by Zhang et al. established a novel function for lactate as a substrate for the generation of lactyl-CoA (la-CoA) to modify lysine residues on histones[2]. Similar to acetylation, succinylation, and crotonylation, the newly-discovered histone lysine lactylation (Kla, L-lactylation) stimulates gene expression in macrophages, providing new insights into the non-metabolic functions of lactate[2]. Subsequent studies have unraveled that Kla influences the differentiation of pluripotent stem cells, promotes the development of cancer[3], and exacerbates microglial dysfunction in Alzheimer's disease[4]. According to a global lacty-lome profiling of a cohort of hepatitis B virus-related hepatocellular carcinoma (HCC), Kla is a ubiquitous modification that preferentially modifies enzymes, and facilitates proliferation and metastasis of HCC cells[5–7].

Now, lactate has been rediscovered as not only a major energy source and gluconeogenic precursor, but also an important signaling

[1]The Province and Ministry Co-sponsored Collaborative Innovation Center for Medical Epigenetics, Key Laboratory of Immune Microenvironment and Disease (Ministry of Education), Tianjin Key Laboratory of Medical Epigenetics, Department of Biochemistry and Molecular Biology, School of Basic Medical Sciences, Tianjin Medical University, Tianjin 300070, China. [2]Frontiers Science Center for Cell Responses, College of Life Sciences, Nankai University, Tianjin 300071, China. [3]Department of Pathology, Tianjin Medical University, Tianjin 300070, China. [4]Department of Cell Biology, School of Basic Medical Sciences, Tianjin Medical University, Tianjin 300070, China. [5]Tianjin Key Laboratory of Retinal Functions and Diseases, Eye Institute and School of Optometry, Tianjin Medical University Eye Hospital, Tianjin Medical University, Tianjin 300070, China. [6]These authors contributed equally: Ziping Niu, Chen Chen, Siyu Wang, Congcong Lu. ✉e-mail: chench@tmu.edu.cn; kzhang@tmu.edu.cn

molecule, serving both metabolic and non-metabolic functions[8]. These findings raise the importance of uncovering the biochemistry involved in the transfer of lactyl moieties to substrates, as well as the removal of lactyl groups. As a relatively new research area, the identity and function of writer and eraser proteins of lactylation are not well-defined. Recent studies suggest that lysine acetyltransferases (KATs) and lysine deacetylases (KDACs), which are responsible for the addition or removal of acetyl groups from histones, may be involved in regulation of lactylation. For example, class I histone deacetylases (HDAC1–3)[9] and Sirt 2[10]–3[11] have been reported as delactylases. Meanwhile, only p300 is identified as potential lactyltransferase in eukaryotes[2]. However, studies have shown that Kla is a widespread modification type in plant fungi[12] and prokaryotes[13] which lack p300 homologs, indicating the presence of more lactylation transferases.

Based on sequence conservation and structural similarity of the catalytic domain, there are three major KAT families: p300/CBP, GNAT, and MYST[14]. Increasing studies have demonstrated that KATs can catalyze a variety of acylations. For instance, p300/CBP has been shown to catalyze almost all types of acylations, including acetylation[15], propionylation, butyrylation[16], crotonylation[17], and succinylation[18]. GCN5 and PCAF from the GNAT family can transfer malonylation and succinylation effectively[19]. HBO1, a member of MYST family, is a versatile histone acyltransferase and can catalyze histone acetylation, propionylation, butyrylation, crotonylation, and benzoylation in vivo and in vitro[20,21]. Our lab recently reported YiaC, a member of the GNAT family, can serve as a lactyltransferase in *E. coli*[13]. Until now, two out of three families of KATs have been determined as lactyltransferases. The findings raise an interesting question of whether MYST KAT family can transfer lactylation.

Here, we report HBO1 as a transferase for lysine lactylation in mammalian cells and it can regulate histone Kla both in vitro and intracellular. Using cell culture-based stable isotope labeling (SILAC) quantitative proteomics, we identify 95 Kla sites that are down-regulated after HBO1-KO, with the majority located on nuclear proteins, specifically histones. Based on SILAC and site-specific antibody validation, we find that HBO1 may preferentially catalyze H3K9la. CUT&Tag assays further demonstrate that HBO1 is required for histone H3K9la on transcription start site (TSS) and the regulated H3K9la can facilitate not only in the important signaling pathways but also in the progress of tumorigenesis. Limited proliferation, migration, and invasion abilities are found in seven different cancer cell lines after HBO1-KO, as well as down-regulated H3K9la. Furthermore, increased levels of H3K9la and HBO1 are observed in clinical cervical cancer tissue samples as compared to normal tissues, suggesting that HBO1's potential role in tumorigenesis through H3K9la-mediated gene regulation. Together, our results reveal that HBO1 regulates histone lactylation and therefore stimulates gene expression.

## Results

### In silico molecular docking predicts lactyl-CoA as the substrate for HBO1

Accumulated evidence has shown that lysine lactylation (L-lactylation) links metabolism and gene regulation[22], and plays an important role in a variety of biological processes, including cancer development[6]. However, the specific mechanism remains unclear. At the same time, limited writers and erasers of lactylation are unrevealed. To date, four delactylases have been reported[9–11], while only p300 shows potential lactyltransferase activity in eukaryotes[2]. Global analysis proves that lysine lactylation is a widespread type of PTM in plant fungi[12] and prokaryotes where there are not p300 homologs, suggesting that more lactylation transferases remain to be discovered. Indeed, we recently revealed that YiaC (a member of GNAT family) functions as a lactyltransferase in the regulation of metabolism in *E. coli*[13]. Until now, two out of three families of lysine acyltransferases (KATs) have been determined as Kla transferases. Compared to p300/CBP and GNAT

family, MYST family, the third family of KAT, has a larger catalytic pocket which gives it an advantage in binding to substrates with higher molecular weight, e.g., lactyl-CoA.

To explore whether MYST family KATs are potential lactyltransferases, we performed in silico molecular docking analysis. Lactyl-CoA, the precursor of Kla, was used as the substrate, while members in MYST family, namely HBO1 (KAT7), MOF (KAT8), TIP60 (KAT5), and MOZ (KAT6A) with known crystal structures in PDB, were selected as potential candidates. Structural analysis shows that all enzymes could accommodate the polar portion of lactyl-CoA (L-Lactyl-CoA) into their hydrophobic pockets. Among the studied enzymes, molecular docking revealed the highest efficient binding affinity (docking energy −15.1 Kcal/Mol) of lactyl-CoA with HBO1 (Fig. 1a), suggesting that HBO1 can better utilize lactyl-CoA as the substrate although distinguished from the binding modes of p300 with la-CoA (Supplementary Fig. 1).

### HBO1 regulates histone lactylation in cells

Previous studies showed that HBO1 is a versatile histone acyltransferase, involved in the catalyzation of acetylation, butyrylation, and crotonylation[20]. Moreover, the KAT activity of HBO1 could be abolished by a mutation of residue 508 glutamic acid to glutamine (E508Q)[23]. Our in silico analysis predicts that lactyl-CoA is a substrate of HBO1, indicating that HBO1 is a potential lactyltransferase.

To investigate its lactylation transfer function, overexpression of HBO1 in HEK293T cells was performed and pan-L-Kla antibody was used to detect the level of lactylation. Meanwhile, overexpression of p300 were done as positive control. Both immunofluorescence (IF) staining and western blotting of histone pan-L-Kla results demonstrated a positive correlation between the expression levels of histone lactylation and HBO1 or p300. We also checked the abundance of acetylation after exogenous overexpression of enzymes, and obtained the similar result. The more abundant the enzymes were, the more lactylation or acetylation was detected (Fig. 1b–d), suggesting HBO1's catalysis activity in lactylation. Next, we determined the key sites of HBO1 in the catalyzation of lactylation. We selected E508 and other five amino acid sites (V472, G485, P510, S512 and R591) that could have the potential for affecting on the interaction between lactyl-CoA and HBO1 by the structural analysis (Supplementary Fig. 1c). Loss-of-function mutations (E508Q, V472A, G485A, P510A, S512A and R591A) were performed on the selected HBO1 amino acid sites and overexpressed in HeLa cells. Subsequently, we examined the level of lactylation and acetylation for these mutations (Supplementary Fig. 2a, b), and found that the E508Q mutation greatly reduced the level of lactylation catalyzed by HBO1, indicating that HBO1 may exert its catalytic activity of Kla by the key lysine residue.

To further determine its histone lactylation transferase activity, HBO1 was knocked down using shRNA, which lead to a decrease of lactylation and acetylation in HeLa cells (Fig. 1e, f) and H460 cells (Supplementary Fig. 2c). We also generated HBO1 knockout clones in HeLa cells using CRISPR-Cas9. Two knockout clones with high efficiency, naming in HBO1-KO-1 (#32) and HBO1-KO-2 (#36), were picked out of 41 clones for downstream experiment (Supplementary Fig. 2d). Single knockout clone cells, as well as mix clone, showed reduced expression of histone lactylation and acetylation after elimination of HBO1 with H3K14ac as a positive control (Given that H3K14ac is a known substrate catalyzed by HBO1)[23], suggesting that HBO1 possesses catalytic activity of histone lactylation transferase (HLT) in cells (Fig. 1g, h).

### HBO1 directly catalyzes histone lactylation in vitro

To further test if HBO1 directly catalyzes histone lactylation, we expressed and purified HBO1 and HBO1 (E508Q) in HEK293T cells (Supplementary Fig. 3a). We then carried out in vitro lactylation assays using core histones isolated from HEK293T cells as substrates, and lactyl-CoA as donors. Acetyl-CoA was also incubated with histone

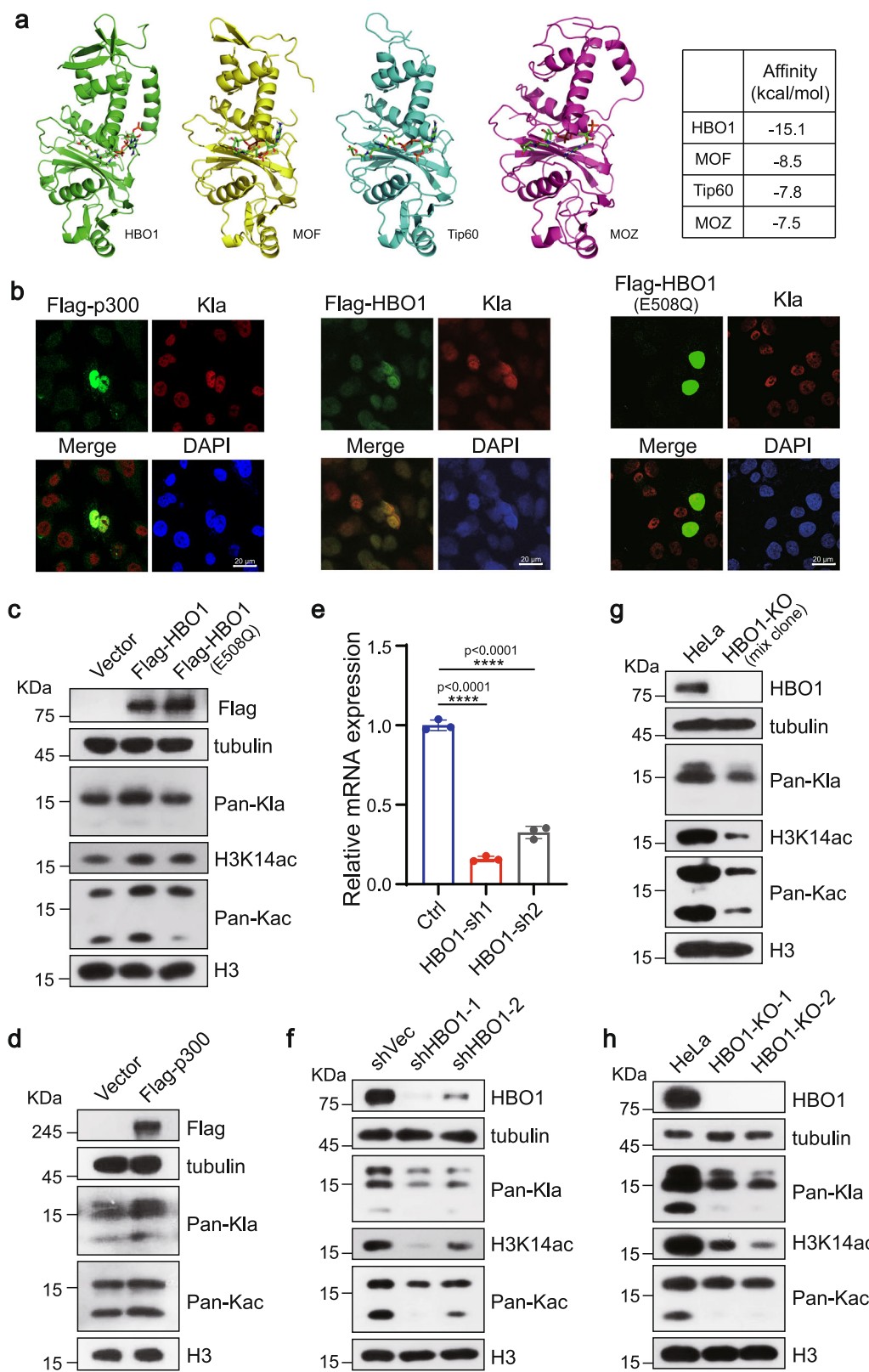

proteins at the present or absence of enzymes (Fig. 2a, b). At the same time, to exclude the influence of HEK293T histone background, we used recombinant histone protein as the reaction substrate to carry out the assay under the same conditions (Fig. 2c, d). The representative results demonstrated that HBO1 can catalyze histone lactylation and acetylation, as expected, HBO1 (E508Q) does not have the catalytic activity.

As previously reported, the acyltransferase domain of HBO1 consists of residues ranging from 336 to 611 aa[20]. We therefore expressed and purified the catalytic domain of HBO1 in *E. coli* (Supplementary Fig. 3b) and incubated it with histones and the corresponding CoA. According to WB results, HBO1 (336-611 aa) can transfer CoA donors to histone substrates (Supplementary Fig. 3c, d). An enzyme dose- and reaction time-dependent increase in total histone

**Fig. 1 | HBO1 regulates lysine lactylation intracellular. a** Predicted structures of HBO1, MOF, Tip60, and MOZ complexes formed with la-CoA, respectively. Proteins are shown with ribbons and binding ligands are shown with ball-and-stick structures. Molecular docking showed that the affinity constant of HBO1 with la-CoA was higher than others. **b** Immunofluorescence (IF) staining showed that exogenous overexpression of wild-type p300 and HBO1 could enhance the lactylation level of HeLa cells, but overexpression of HBO1 (E508Q) could not. Scale, 20 μm. **c**, **d** Western blotting analysis showed HBO1 but not HBO1 (E508Q) in HEK293T cells increased histone lactylation levels (**c**). Overexpression of p300 was conducted as positive control (**d**). **e** The bar chart shows the relative mRNA expression level of HBO1 gene after HBO1 knockdown. All data (mean ± SD, two-tailed Student's *t*-test) were from three biological replicates ($n = 3$). *P*-values are indicated in the figure. $p < 0.0001$ (****). **f** Western blotting analysis showed transient HBO1 knockdown by lentiviral-based shRNAs leads to downregulation of histone lactylation in HeLa cells. **g**, **h** Knockout (KO) of HBO1 protein leads to downregulation of histone lactylation levels. Mixed cloned cells with HBO1 knockout were obtained 48 h after lentivirus infection. Two stable clone cells were screened from 41 clone cells. All immunoblots and immunofluorescence staining had three biological repetitions, with similar results. Source data are provided as a Source Data file.

lactylation was observed, confirming the lactyltransferase activity of HBO1 (Fig. 2e, f). Additionally, isothermal titration calorimetry (ITC) was used to measure the interactions of HBO1 (336-611) with CoA. ITC data showed that the Kd value of binding between HBO1 and lactyl-CoA was 0.52 μM whereas HBO1 and ac-CoA was 1.46 μM (Fig. 2g, h, Supplementary Data 1), suggesting that HBO1 can catalyze lactylation in vitro.

## Quantitative proteomics analysis reveals that nuclear proteins are the major Kla substrates of HBO1

Both in cells and in vitro assays prove HBO1 could catalyze histone Kla. Previous studies have demonstrated that Kla occurs on both histone and non-histone proteins[6]. In consistent, our WB data confirmed the widespread distribution of lactylation throughout the cell and revealed that HBO1 regulates lactylation since the global Kla was reduced after HBO1-KO. (Supplementary Fig. 4).

To determine the endogenous Kla substrates of HBO1, we performed a SILAC-based quantitative proteomics analysis of global lactylation in wild-type and HBO1-KO HeLa cells (Supplementary Fig. 5a). In total, 451 Kla sites were identified in 262 proteins (Supplementary Fig. 5b, Supplementary Data 2), with 37% of proteins carrying multiple Kla sites (Supplementary Fig. 5c). Sequence characterization indicated that HBO1 tended to catalyze Kla on characteristic amino acid sequences (Supplementary Fig. 5d, e). GO annotation analysis revealed that the majority of lactylation events occurred in the cell nucleus and were involved in chromatin remodeling and assembly (Supplementary Fig. 5f, g). After normalized the Kla abundances to the expression of their corresponding proteins, 95 Kla sites (10%) were found to be 1.5-fold downregulated in HBO1-KO cells (Fig. 3a, b). In agreement with our WB results (Supplementary Fig. 4d), GO enrichment indicates that the regulated Kla are engaged in chromatin remodeling and protein-DNA complex organization (Fig. 3c). Compared to cadherin binding for global Kla proteins, more HBO1 substrate candidates have molecular functions linked to nucleosomal DNA and nucleosome binding, suggesting that nuclear proteins are the major substrates of Kla catalyzed by HBO1.

## HBO1 preferentially catalyzes histone H3K9la

Considering that the major Kla substrates of HBO1 localize in the cell nucleus and the essential roles of histones in epigenetic regulation, we isolated histones before L-Kla enrichment and investigated the abundance changes of Kla sites after HBO1-KO in HeLa cells (Supplementary Fig. 5a). Notably, 72 histone Kla sites were quantified among canonical histones and histone variants, 29 of which were 1.5-fold decreased in HBO1-KO cells (Fig. 3d, Supplementary Fig. 6a, Supplementary Data 3). Representative MS/MS spectra of H3K9la (Fig. 3e) and other Kla sites (Supplementary Fig. 6b) and their corresponding Kac sites (Supplementary Fig. 6c) with high confidence illustrate the reliability of our MS dataset.

The amount of metabolite could affect histone acylation expression even in the absence of acyltransferase[24]. Therefore, we added extracellular lactate with and without the presence of HBO1 and demonstrated that histone Kla was downregulated after shHBO1 knockdown. As predicted, Kla was upregulated upon the addition of lactate (Supplementary Fig. 7a). Similar results were found in H460 cells (Supplementary Fig. 7b), suggesting that histone Kla sites might occur in both HBO1-dependent and -independent ways. Site-specific Kla antibodies were then utilized to confirm the changes in multiple histone Kla expression levels after CRISPR-Cas9-knockout or shRNA knockdown of HBO1. Validated by site-specific antibodies, a reduction in histone lactylation was observed on the representative histone H3 and H4 sites, including H3K9, H3K14, H3K18, H4K5, H4K8, H4K12, and H4K16, which was consistent with the MS data (Fig. 4a–c). Additionally, we verified the catalytic effect of HBO1 on histone lactylation through a group of experiments including HBO1 overexpression and knockdown. Besides, we conducted a rescue experiment of HBO1 in HBO1-KO HeLa cells, which further confirmed the direct catalytic effect of HBO1 on histone sites, especially H3K9la (Fig. 4d).

WM-3835 is a known HBO1 inhibitor that binds to the lactyl-CoA binding site competitively (Supplementary Fig. 7d). Treatment with WM-3835 resulted in a dose- and time-dependent decrease in overall histone lactylation, where histone H3K9la were more sensitive to HBO1 inhibition (Fig. 4e, f). We then investigated the changes in histone Kla after adding extracellular lactate. As predicted, higher expression levels of Kla were observed upon addition of lactate. While extra lactate did not rescue H3K9la in HBO1-KO cells, implying that its lactylation might require the presence of HBO1 (Fig. 4g). We also conducted in vitro experiments and confirmed that HBO1 can indeed lactylate H3K9 (Supplementary Fig. 7c). Given that abundance of H3K9la was decreased most significantly after knockout of HBO1, we focused on the specific regulation of HBO1 to H3K9la.

## The scaffold proteins promote the activity of HBO1 as lactyltransferase

Accumulated evidence has shown that scaffold proteins play critical roles in maintaining enzyme activity[20]. There are two types of known scaffold proteins of HBO1, namely BRPF1/2/3 and JADE1/2/3. According to previous research, BRPF2 and JADE1 have relatively high activity in promoting HBO1 acyltransferase function[20] and are therefore selected as representative scaffolds to investigate their abilities in regulating HBO1 ability as a lactyltransferase.

In accordance with earlier reports[20], Kac was downregulated after shRNA knockdown of BRPF2 or JADE1 (Fig. 5a, c, Supplementary Fig. 8a, b). Similar decreases in Kla and H3K9la expression levels were detected, indicating that both proteins have the capability to promote HBO1 activity. More reductions were observed when JADE1 was knocked down. We then co-overexpressed HBO1 and JADE1. Compared to HBO1-overexpression alone, co-overexpression of HBO1-JADE1 complex resulted in increased Kla, but not Kac, conforming that JADE1 may act as a promoter in regulating HBO1 Kla activity (Fig. 5b). Next, we performed in vitro acylation assays using core histones as substrates. Flag-HBO1 alone and HBO1-BRPF2, HBO1-JADE1 complexes were purified (Supplementary Fig. 8c, d) and incubated with substrates, respectively. An enhanced ability to catalyze histone Kla (Fig. 5d, f), as well as Kac (Fig. 5e, g), were found when HBO1 was present as a complex, demonstrating that scaffold proteins can promote HBO1 Kla catalytic activity, especially H3K9la.

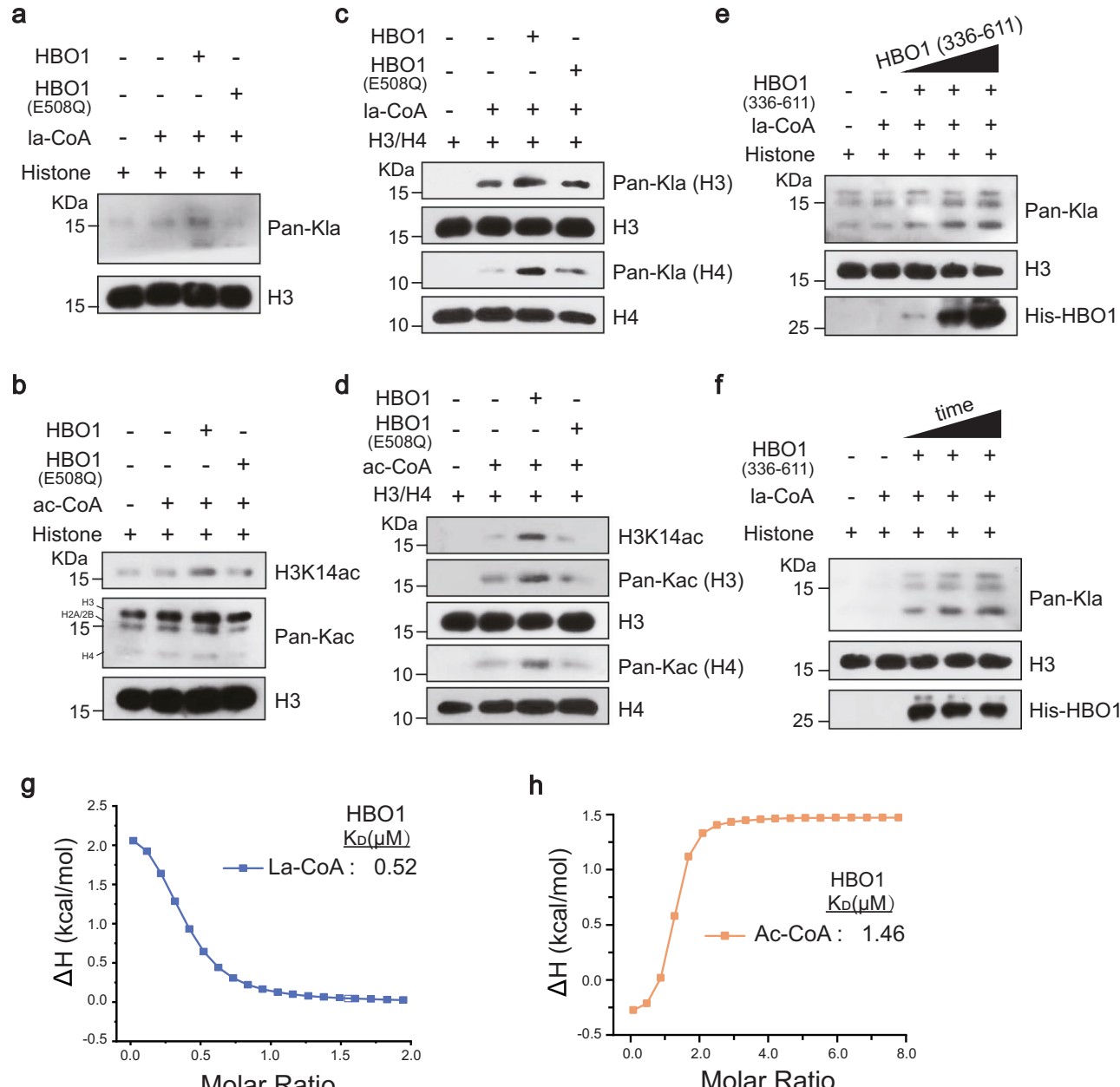

**Fig. 2 | HBO1 directly catalyzes lactylation in vitro. a, b** HBO1 instead of HBO1 (E508Q) enhanced the lactylation level of histones in vitro. HBO1 and HBO1 (E508Q) were overexpressed in HEK293T cells and affinity purified using anti-Flag M2 beads. The purified proteins were incubated with la-CoA (**a**) or ac-CoA (**b**) and histone extracted from HEK293T as the reaction substrate for in vitro histone acylation assay. **c, d** HBO1 has catalytic activity for the lactylation of recombinant protein H3/H4 but not HBO1 (E508Q) in vitro. HBO1 or HBO1 (E508Q) were incubated with lactyl-CoA (**c**) or acetyl-CoA (**d**) and recombinant protein H3/H4 as the reaction substrates, respectively, for acylation assay in vitro. **e, f** The lactylation modification level of histones catalyzed by HBO1 (336-611) were enzyme dose-dependent (**e**) and reaction time-dependent (**f**). HBO1 (336-611) was purified from *E.coli*. Reaction time points: 10, 30 and 60 min. Enzyme does: 0.5, 1.5, and 2.5 μg. **g, h** Isothermal titration calorimetry (ITC) analysis of the affinity of recombinant HBO1 (336-611) with lactyl-CoA (**e**) and acetyl-CoA (**f**), respectively. All immunoblots and isothermal titration calorimetry (ITC) analysis had three biological repetitions, with similar results. Source data are provided as a Source Data file.

## HBO1 is required for histone H3K9la on TSSs

To determine how HBO1 contributes to histone H3K9la, we performed genome-wide Cleavage Under Targets and Tagmentation (CUT&Tag) followed by high-throughput DNA sequencing for HBO1 and H3K9la. A high reproducibility of CUT&Tag dataset was obtained with Pearson correlation coefficients over 0.9 (Supplementary Fig. 9a). In all, 11524 genes were found to be overlapping, and 42.7% of these genes are within 3 kb range of TSSs, suggesting that the peaks of H3K9la and HBO1 were co-located and heavily enriched in the TSS region (Fig. 6a–c). We then use CUT&Tag technology for H3K9la in both WT and HBO1-KO HeLa cells. In total, 17062 binding peaks were found to be downregulated in HBO1-KO Hela cells, of which 49.2% were mapped to TSSs and 68.0% of them to regions within 3 kb of TSSs (Fig. 6d). The differential binding peak of H3K9la was significantly higher in HeLa WT than in HBO1-KO cells within the range of ±3 kb in the TSS region (Fig. 6e). Thus, a significant reduction of TSSs were detected in HBO1-KO cells, and the reduction of gene promoters, such as *AQP1, LAMC2,* and *F10*, was further confirmed by qPCR in genome and transcriptome levels (Fig. 6f–h, Supplementary Fig. 9b, c), suggesting that HBO1 is required for histone H3K9la on TSSs.

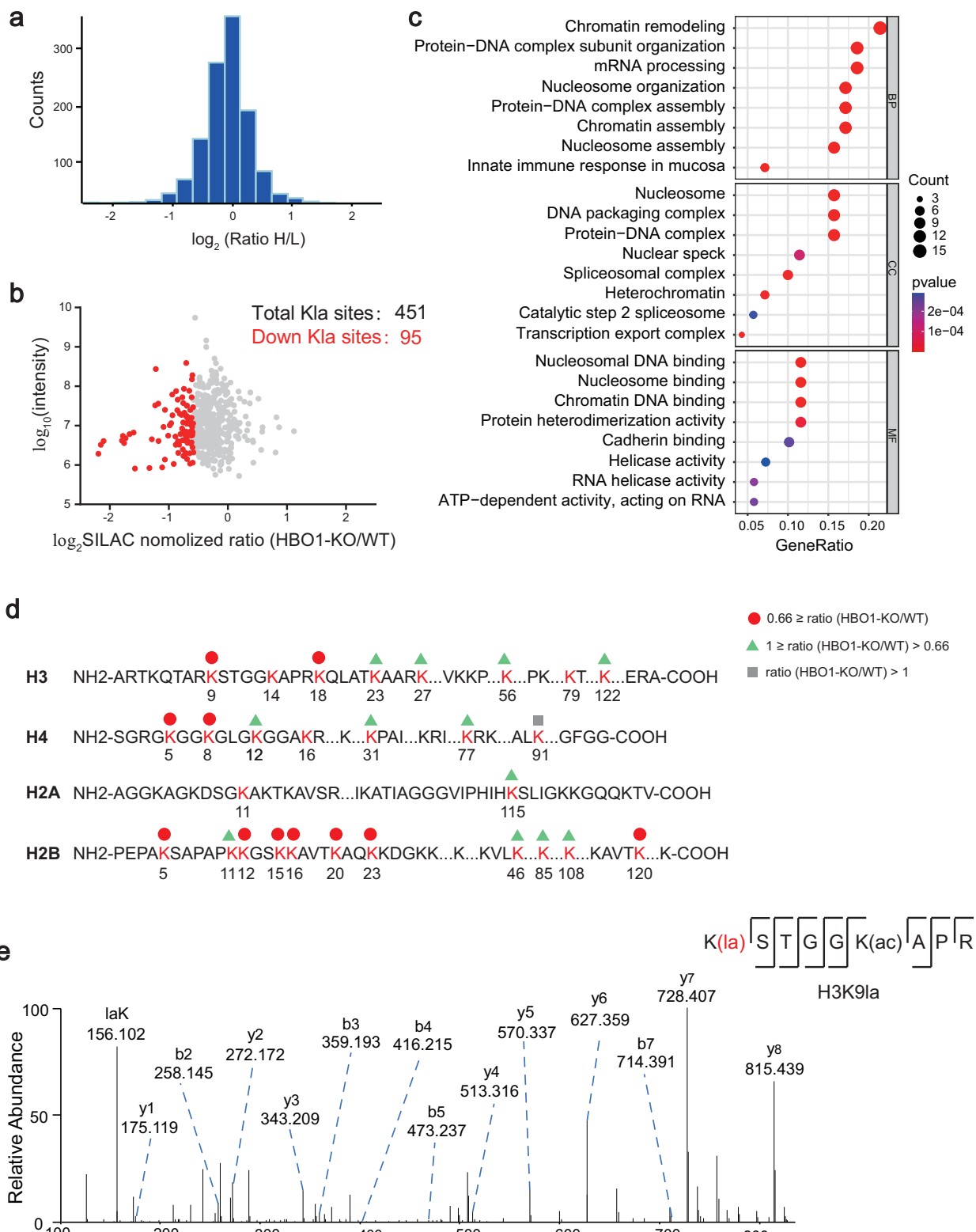

**Fig. 3 | Identification of endogenous Kla substrates of HBO1. a** The histogram showed experimentally determined relative protein abundance distributions for the samples used to analyze lactylation. **b** Scatterplot showed the ratio of Kla peptides in HBO1-KO HeLa cells versus wide-type HeLa cells (normalized by protein abundance). **c** GO analysis of down-regulated lactylated proteins after HBO1 knockout. Over-representation test was used to calculate GO term enrichment with FDR for multiple test correction. The p-value cutoff = 0.05 and q-value cutoff = 0.1 were selected as the cutoff criteria. Benjamini and Hochberg correction was used to adjust p-values. **d** Illustration of histone Kla sites regulated by HBO1 identified in HeLa cells. **e** The representative MS/MS spectrogram of H3K9la peptide (K(la) STGGK(ac)APR).

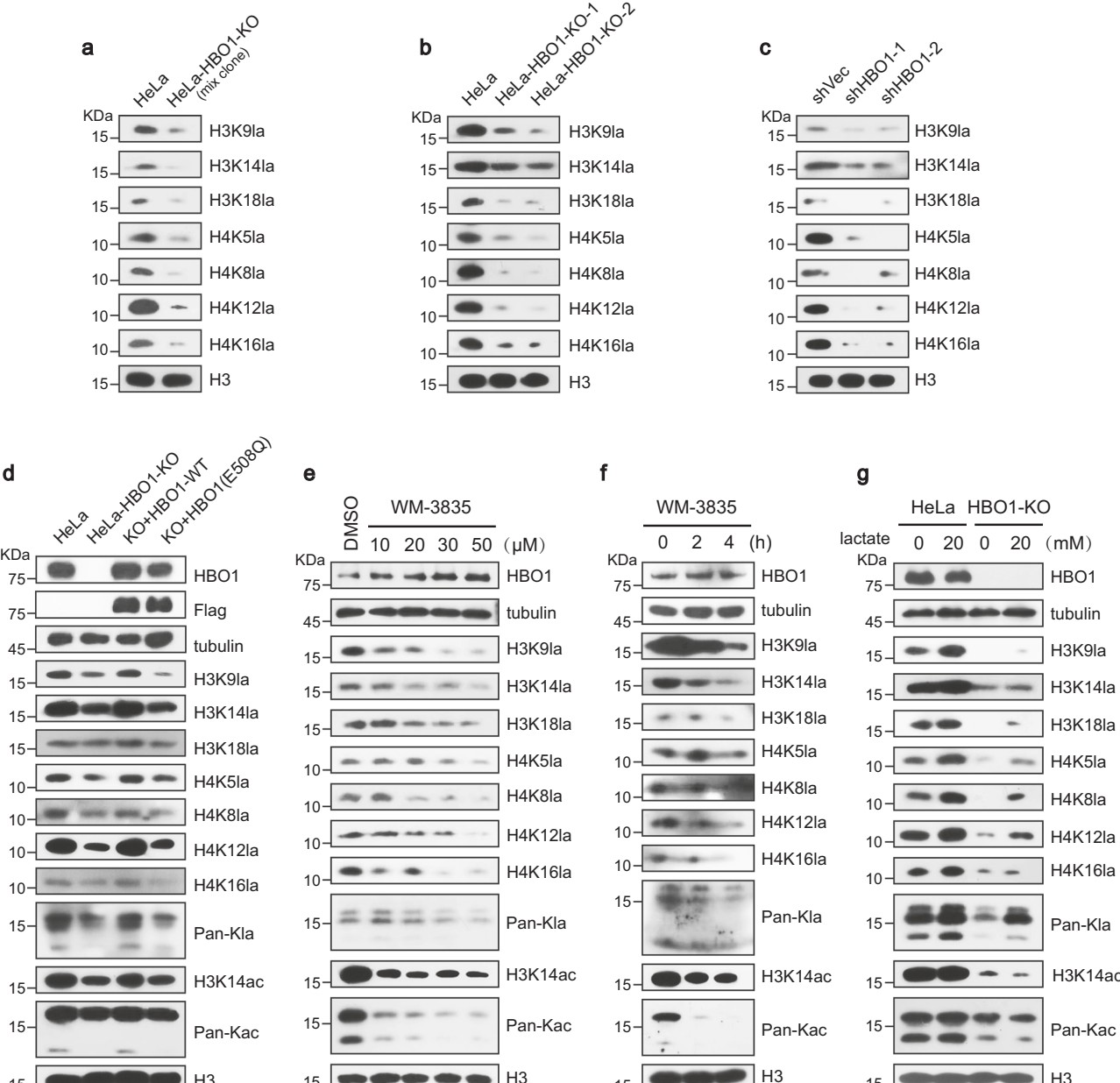

**Fig. 4 | Characteristics of histone lactylation sites regulated by HBO1.**
**a–c** Western blotting analysis of changes for various histone lactylation sites after HBO1 knockout in mixed clone (**a**) and stable clones (**b**), and HBO1 knockdown by lentiviral-based shRNAs (**c**). **d** Western blotting showed that re-expression of HBO1-WT instead of HBO1 (E508Q) in HBO1-KO cells could rescue the level of histone lactylation. **e, f** Western blotting analysis showed the downregulation of lactylation levels in global histones and various histone sites was WM-3835 dose-and time-dependent. WM-3835: a specific inhibitor of HBO1. **g** Lactylation of histones is lactate-dependent mediated by HBO1. HeLa cells and HBO1-KO HeLa cells were stimulated with extra addition of 0 mM or 20 mM sodium lactate, respectively. All immunoblots had three biological repetitions, with similar results. Source data are provided as a Source Data file.

Given that HBO1 can also catalyze histone acetylation, particularly H3K14ac, we performed a set of comparative experiments to further verify the Kla-dependent gene regulation. Lactic acid (LA) has been considered as the precursor source of lactyl-CoA, which could be transferred to substrates by lactyltransferase. We used oxamate to inhibit LDHA activity, which could lead to decrease of LA concentration in cells, as well as added sodium lactate externally to increase LA concentration in cells. As expected, histone lactylation but not acetylation was affected by adding oxamate or sodium lactate, respectively (Supplementary Fig. 10a, d). Subsequently, *AQP1*, *LAMC2*, and *F10* genes were analyzed by QPCR of H3K9la or H3K14ac, and the results revealed that H3K9la was significantly enriched in the TSSs (Supplementary Fig. 10b, c, e, f).

## HBO1 regulates H3K9la and promotes malignant behavior of cancer cells

Next, GO and KEGG pathway analysis was performed for the 17,062 down-regulated genes targeted by H3K9la in HBO1-KO cells (Supplementary Fig. 9d, e). Interestingly, pathways involved in carcinogenic mechanisms were enriched, e.g., PI3K-AKT signaling, MPAK signaling, Wnt signaling, and Hippo signaling, suggesting that HBO1-mediated H3K9la may activate the transcription of genes encoding known tumorigenesis.

To explore the potential effect of HBO1-mediated H3K9la on the malignant behaviors of cancer cells, we constructed the 7 types of different HBO1-KO cell lines, including HeLa, HepG2 (hepatocellular carcinoma), U87MG (gliomas), KYSE-30 (esophageal squamous cell

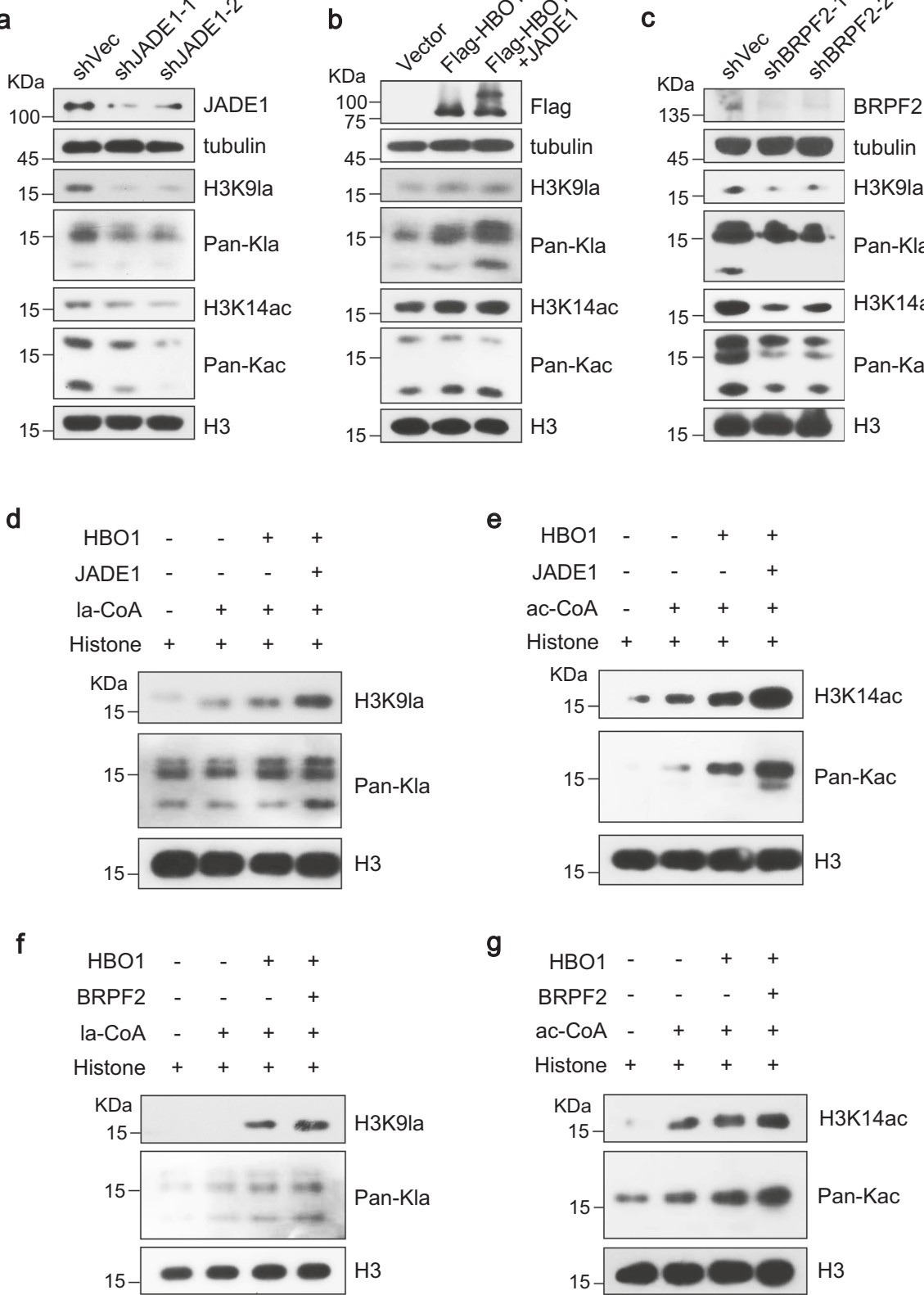

**Fig. 5 | The scaffold proteins promote HBO1 lactyltransferase ability intracellular and in vitro. a**–**c** Western blotting analysis of histone lactylation after lentiviral-based shRNAs transient knockdown of JADE1 (**a**) or BRPE2 (**c**), or overexpression of HBO1 alone and HBO1-JADE1 complex (**b**). **d**–**g** Scaffold protein enhances the catalytic activity of HBO1 for histone lactylation in vitro. Flag-HBO1,

Flag-HBO1-JADE1 or Flag-HBO1-BRPF2 complexes were purified from HEK293T cells by immunoaffinity of anti-Flag M2 beads, and incubated with la-CoA (**d**, **f**) and ac-CoA (**e**, **g**) respectively for in vitro histone acylation assay. All immunoblots had three biological repetitions, with similar results. Source data are provided as a Source Data file.

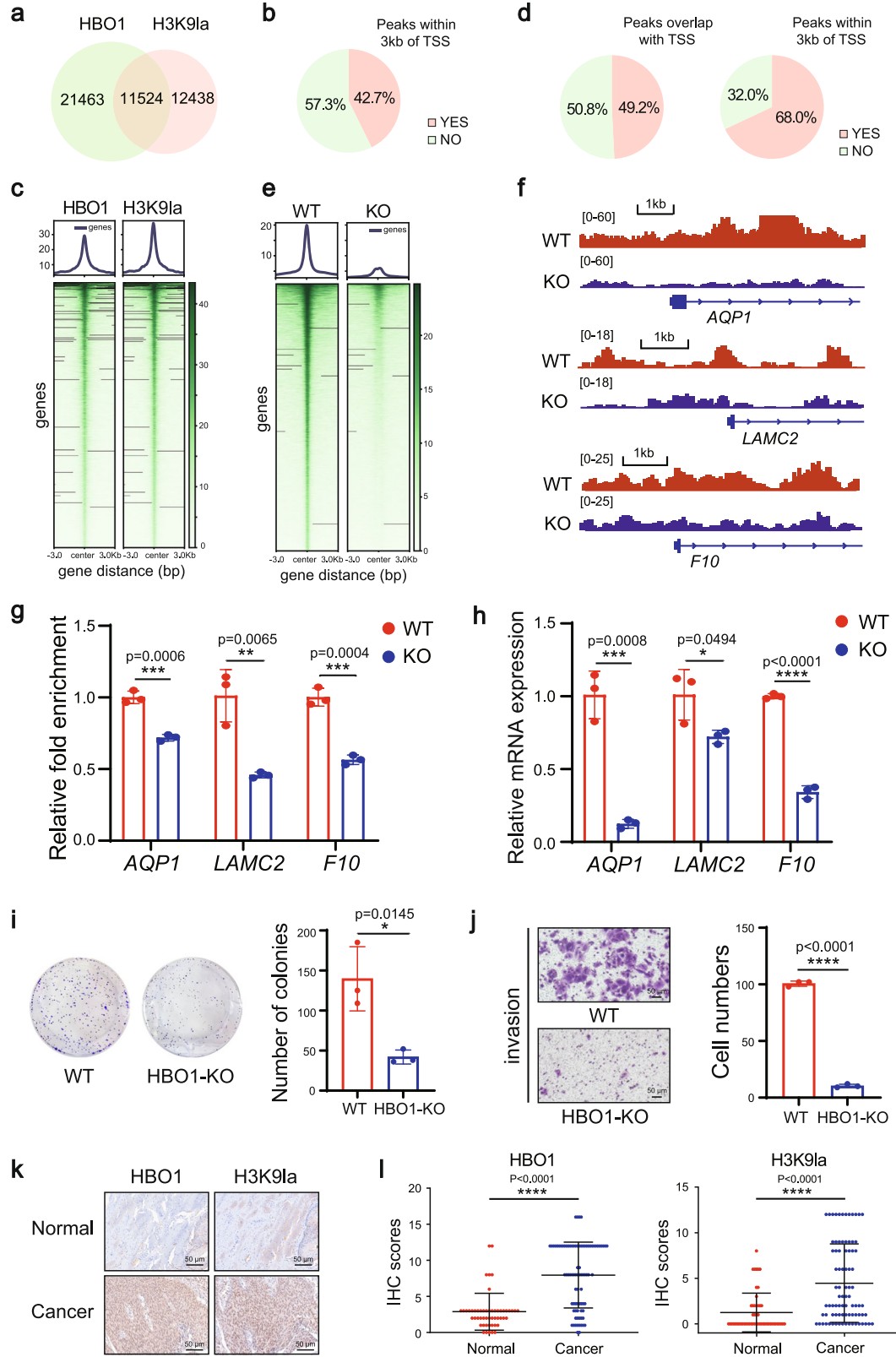

carcinoma), MDA-MB-231 (breast carcinoma), HCT116 (colon cancer), and H460 (non-small cell lung cancer). As expected, HBO1-KO lead to the decrease of H3K9la level in these cells. We further conducted a set of experiments including clone formation experiment, scratch experiment, and transwell invasion experiment. Compared to WT, HBO1-KO resulted in the inhibition of cell proliferation, migration, and invasion (Fig. 6i, j, Supplementary Fig. 11a–t), indicating that HBO1-

mediated activation of H3K9la pathways is associated with tumorigenesis.

To further investigate the significance of HBO1-mediated H3K9la, we examined the expression levels of HBO1 and H3K9la in clinical tissues including 84 cervical cancer cell samples and 52 normal clinical cervical samples. We found that the levels of HBO1 and H3K9la in cervical cancer tissues were obviously higher than those in normal

**Fig. 6 | Genomic analysis of transcriptional consequences of HBO1 and H3K9la.**
**a** Venn diagram showed co-localization analysis of HBO1 and H3K9la peaks in HeLa cells. **b** Pie chart showed that 42.7% of 11524 peaks co-located by HBO1 and H3K9la were found within 3 kb of the TSS region. **c** Heat maps generated by CUT&Tag data analysis illustrate the co-located peaks of HBO1 and H3K9la within ±3 kb in the TSS region. **d** The pie chart showed that 49.2% of differential peaks of enrichment overlap in the TSS region in HeLa and HBO1-KO cells, and 68.0% of them were distributed within ±3 kb in the TSS region. **e** The binding density of H3K9la visualized by deepTools: Heat maps show differential binding peaks of H3K9la distributed within ±3 kb in the TSS region in HeLa and HBO1-KO cells. **f** Representative traces of CUT&Tag showed that H3K9la was enriched in the TSS region of *AQP1*, *LAMC2* and *F10* genes. The red and blue tracks represent the peak of WT and HBO1-KO cell enrichment, respectively. **g** CUT&Tag results for *AQP1*, *LAMC2* and *F10*

genes were verified by qPCR ($n$ = 3 biological repetitions). **h** RT-qPCR analysis showed the transcription levels of *AQP1*, *LAMC2* and *F10* genes in HeLa and HBO1-KO cells ($n$ = 3 biological repetitions). **i, j** Cell clonal formation assay and transwell invasion assay showed inhibition of proliferation (**i**) and invasion ability (**j**) (Scale, 50 μm) of HeLa cells after HBO1 knockout, respectively ($n$ = 3 biological repetitions). **k** HBO1 expression and H3K9la level were detected with immunohistochemistry (IHC) in a tissue microarray. Typical images of IHC are shown. Scale, 50 μm. **l** The level of H3K9la and the expression level of HBO1 in cervical tumors were significantly higher than that in normal cervical tissues (patients with cervical cancer $n$ = 84, normal $n$ = 52). Data are presented as mean ± SD, two-tailed Student's *t*-test. *P*-values are indicated in the figure. Source data are provided as a Source Data file.

cervical tissues (Fig. 6k, l). Moreover, the expression of HBO1 was positively correlated with H3K9la (Supplementary Fig. 11u), suggesting that HBO1-mediated H3K9la may promote the cervical tumorigenesis. Together, our data indicates that HBO1 was directly involved in the H3K9la of TSSs, regulating gene transcription and promoting tumorigenesis (Fig. 6).

## Discussion

Multiple global lactylome profiles have demonstrated that lysine lactylation is a ubiquitous modification type, ever since its discovery in 2019. Similar to other histone lysine acylations, such as acetylation, succinylation, and crotonylation, histone lactylation has been found to be involved in epigenetic regulation[25–29]. However, lactylation is distinct in that it directly links metabolism and gene regulation, as lactate is the end product of glucose metabolism and the main fuel for the TCA cycle. To gain a deeper understanding of the molecular functions of lactylation, it is important to identify the enzymes responsible for the transfer as well as the removal of lactyl groups. In this study, we revealed that HBO1 functions as a lactyltransferase and can catalyze lactylation of histones both in vitro and intracellular, thereby mediates gene expression (Fig. 7).

HBO1, also known as KAT7 or MYST2, is a member of the highly conserved MYST KAT family. It has previously been reported to function as a major transcriptional regulator, primarily via histone H3K14 acetylation[30]. Recent efforts have shown that HBO1 is a versatile histone acyltransferase that catalyzes not only acetylation but also crotonylation, propionylation and butyrylation[20]. To explore the possibility of HBO1 acting as a lactyltransferase, we firstly conducted in silico molecular docking, which predicted HBO1 can bind lactyl-CoA with higher biding affinity compared to other MYST family members (Fig. 1a). Then, we overexpressed HBO1 in HeLa and HEK293T cells and found corresponding increased Kla levels (Fig. 1b, c). In consistence, HBO1 knockdown and knockout assays resulted in lower lactylation (Fig. 1e, h). Subsequently, we incubated HBO1 with isolated histones or recombinant histone H3/H4 and lactyl-CoA as substrates. An enzyme dose and reaction time-dependent increase of Kla were observed (Fig. 2), indicating that HBO1 can directly catalyzes histone lactylation.

To precisely identify the Kla sites regulated by HBO1, we coupled genetic deletion of HBO1 in HeLa cells with SILAC-based quantitative mass spectrometry. Global lactylome profiling showed 95 Kla sites were 1.5-fold downregulated after HBO1-KO, where majority of lactylation events occurred in the cell nucleus (Fig. 3). Therefore, we purified histones and found 29 Kla down-regulated sites where H3K9la were identified with the most significant difference. Site-specific histone Kla antibodies were used to validate the histone lactylation changes regulated by HBO1. These data clearly demonstrate that histone H3K9la is mediated by HBO1 (Fig. 4). Similar to most histone HATs, HBO1 can interact with distinct scaffold proteins to form functionally different catalytically active complex[31,32]. Therefore, we assessed the functional effect of known members of the HBO1 complex. JADE1-knockdown decreased Kla whereas JADE1-overexpression

increased Kla, suggesting the lactyltransferase ability of HBO1 is promoted by JADE1. Similar results were found with scaffold protein BRPF2 (Fig. 5).

To further understand the role of HBO1 in regulating gene expression, we performed CUT&Tag with sequencing analysis for HBO1 and H3K9la. 11524 genes were found to be co-located and 42.7% of these genes were enriched in the TSS region. Although our data clearly demonstrate that HBO1 is a predominant enriched in the TSSs and contributes to catalyze the histone lactylation, how HBO1 regulates transcription remains unclear. Therefore, we used CUT&Tag for H3K9la in both WT and HBO1-KO HeLa cells, resulting a total of 17062 down-regulated binding peaks after HBO1-KO. GO analysis indicates that regulated genes are enriched in pathways associated with tumorigenesis We further showed that HBO1-KO caused the down-regulation of H3K9la and inhibited malignant behaviors of tumor cells including proliferation, migration, and invasion. Meanwhile, we found that HBO1 expression and H3K9la levels were higher in cervical cancer clinical samples than in normal cervical tissues, suggesting that HBO1-mediated H3K9la promotes cervical tumorigenesis. Together, these results indicate that HBO1 is highly enriched in TSSs of actively transcribed genes, and may directly involve in the H3K9la-mediated gene regulation and promote tumorigenesis.

It is worth noting that we identified several key genes such as *AQP1*, *LAMC2*, and *F10* regulated by HBO1-mediated-histone H3K9la. In our data, the regulation of *AQP1*, *LAMC2*, and *F10* by H3K9la has also been further verified by QPCR through the addition of oxamate and sodium lactate. Studies have shown that *AQP1* is a key target of local invasion of breast cancer, involving in migration and invasion of breast cancer cells, and therefore targeting *AQP1* offers promise for breast cancer treatment[33]. Besides, *LAMC2* can induce the development and differentiation of pancreatic cancer cells and drive tumor cell invasion, migration, and pellet formation, and thus it is critical for tumor growth and metastasis as a potential therapeutic target and/or biomarker[34]. Additionally, *F10* derived from bone marrow is thought to promote tumor immune escape through proteinase-activated receptor 2 signaling[35]. Together, it suggests that HBO1 holds the potential of affecting the expression of oncogenic genes by regulating H3K9la, and the development of drugs like WM-3835 plays a potential role in inhibiting tumor growth and migration.

In summary, we report that HBO1 is a lactyltransferase and mediates gene transcription through catalyzing histone H3K9la. Our results support the emerging theme that histone lactylation can play a critical role in epigenetic regulation and possible other biological processes. Numerous studies have proven that dysregulation of histone acylation could lead to various diseases including cancers[36–38]. The expanding knowledge of regulatory enzymes is fundamental to uncover the roles in physiology and pathology of the newly-discovered PTM, providing potential targets for drug therapy. Our study provides compelling evidence that HBO1 is highly enriched in the TSSs region and contributes to histone lactylation, suggesting that HBO1-mediated H3K9la may activate the transcription of genes encoding tumorigenesis.

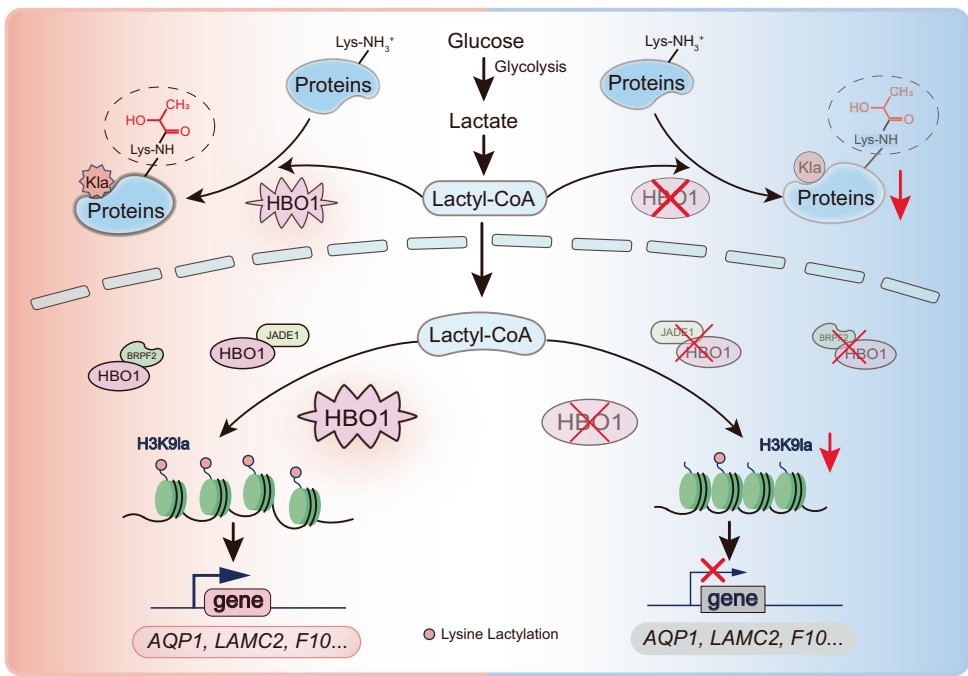

**Fig. 7 | Graphic model as discussed in the text.** HBO1 catalyzes lysine lactylation and mediates histone H3K9la to regulate gene transcription.

## Methods

### Ethics statement
Cervical tumor tissue microarray ZL-Utrsur1601 was purchased from WELLBI. Our study complies with all relevant ethical regulations and was approved by the Ethics Committee of Shanghai Zhuoli Biotech Company (Shanghai, China) (Ethics number: ZLL-15-01).

### Cell culture
HeLa (CL-0101), HEK293T (CL-0005), HepG2 (CL-0103), U87MG (CL-0238), MDA-MB-231 (CL-0150B) and HCT116 (CL-0096) cells were cultured with DMEM (VivaCell) containing 10% fetal bovine serum (FBS) (VivaCell) and 1% penicillin-streptomycin (Thermo Fisher Scientific), and H460 (CL-0299), and KYSE-30 (CL-0577) cells were cultured with RPMI 1640 (VivaCell) containing 10% FBS and 1% penicillin-streptomycin. These cell lines were purchased from Procell Life Science&Technology Co., Ltd. All cells were grown in a 37 °C, 5% $CO_2$ incubator.

### Western blot analysis
Protein samples were separated with 12% or 15% SDS-PAGE gel and then transferred to a nitrocellulose filter membrane, which was blocked with 5% BSA at room temperature for 1 h and incubated with primary antibody overnight at 4 °C. TBST (20 mM Tris-HCl pH 7.6, 150 mM NaCl, and 0.1% Tween 20) washed 3 times and incubated the second antibody for 2 h at room temperature. Next, the membrane was washed 3 times with TBST. Finally, the membrane was exposed to X-ray film in a darkroom. All uncropped and unprocessed scans relevant in this paper are available in the Source Data file.

### Immunofluorescence staining
HeLa cells were washed with PBS 3 times in the cover glass and fixed at room temperature with 4% PFA for 30 min. 0.5% Triton X-100 was incubated in permeable cells at room temperature for 15 min and blocked with 5% BSA for 1 h. The cells were incubated in 4 °C overnight with the corresponding primary antibody. After washing with PBS 3 times, fluorescent secondary antibody was incubated at room temperature for 1 h. The cells were incubated with fluorescent mounting medium with DAPI at room temperature for 5 min for staining and sealing. The slides were observed under an Axio-Imager_LSM-800 microscope.

### Molecular docking
The structure of lactyl-CoA (CHEBI ID: 15529) was retrieved in SMILES format from the ChEBI [https://www.ebi.ac.uk/chebi/] database. The format of the ligands was then converted into PDB format using the software Chem3D for further molecular docking. PDB was used to download the receptor protein. Here, HBO1 (PDB: 6MAJ)[23], TIP60 (PDB: 2OU2) MOZ (PDB: 2RC4)[39], MOF (PDB: 2GIV) and p300 (PDB: 6GYR)[40] were taken as the target receptor proteins. The proteins were subjected to the AutoDock Tool 4.2[41,42]. To perform the steps of protein preparation. AutoDock Vina Software was used in our in silico study, to perform molecular docking of ligand with proteins. AutoDock Vina comes with the feature of the calculation of grid maps automatically. After the successful docking, confirmation of the binding position of ligand into the receptor proteins and calculation of bond distance has been done by PyMol [www.pymol.org] and LigPlus. PyMol and Ligplot[43] allow the clear visualization of binding of ligand-protein with its polar bonds as well as bond distance and hydrophobic interactions.

### ITC measurement
The experiments were performed at 25 °C on a MicroCal PEAQ-ITC isothermal titration calorimeter (Malvern Instruments). The reaction cell containing 50 µM HBO1 was titrated with 500 µM different acetyl-CoA. The volume of the first injection was 0.5 µl of 500 µM acetyl-CoA, and the following 18 injections were 2.0 µl. The binding isotherm was fit with the Origin 8.0 software (OriginLab).

### Cell clonal formation assays
The cells to be observed were counted and 1000 cells/well were inoculated into the 6-well plate culture plate and the medium was changed every 3 days. About a week after cell culture, the cells were washed twice with PBS and fixed with 1 ml 4% paraformaldehyde for 30 min. The cells were stained with crystal violet solution for 5–10 min, and the excess crystal violet solution was washed away with PBS. The formation of cell clones was photographed. The results were analyzed by Image J software (v. Image J2).

## Wound healing assays

After the desired cells were monolayered on a 6-well plate, the cell layer was scratched with a sterile pipette tip. Cells were replaced with fresh serum-free medium and cultured in a 37 °C 5% $CO_2$ incubator. At 0 h, 24 h, and 48 h after scratching, the cells at the same location were recorded by microscope. The results were analyzed by Image J software (v. Image J2). Each experiment was repeated at least three times.

## Transwell invasion assay

The Matrigel was diluted with FBS-free medium at an appropriate proportion and spread into a transwell chamber with a pore size of 8.0 μm, then placed in the incubator at 37 °C until the Matrigel solidified. The cells of interest were suspended on FBS-free medium and spread to the upper chamber of the chamber, and complete medium was added to the lower chamber. The invaded cells were fixed with 4% paraformaldehyde and stained with 0.5% crystal violet. Microscopically recorded the number of cells invaded. Each experiment was repeated at least three times.

## Immunohistochemistry

Cervical tumor tissue microarray ZL-Utrsur1601 was purchased from WELLBI. The tissue sections were de-paraffinized in xylene, rehydrated, and boiled for 10 min in antigen retrieval buffer (TE buffer, pH 9.0). After retrieval, the sections were washed with distilled water and endogenous peroxidase activity was blocked using 3% $H_2O_2$ for 15 min and then blocked with blocking solution (10% goat serum in PBS). Samples were incubated with primary antibodies overnight at 4 °C, washed three times with PBST buffer, and then incubated with anti-rabbit IgG. Sections were counterstained with hematoxylin. Signals of immunohistochemistry data in tumor cells were visually quantified using a scoring system from 1 to 3, multiplied intensity of signal and percentage of positive cells (signal: 0 = no signal, 1 = weak signal, 2 = intermediate signal, and 3 = strong signal; percentage: 10–100%).

## Purification of recombinant proteins

Plasmid HBO1 (336-611)-pET28a was transferred into *E. coil* BL21 (DE3) and shaken in LB medium containing kanamycin (50 μg/ml) at 37 °C shaker until $OD_{600}$ value reached 0.6-0.8. 0.5 mM IPTG was added and the culture was continued incubating overnight at 16 °C. Lysis of *E. coil* was done by ultraphonic and then centrifugation at 16000 g at 4 °C for 30 min. The supernatant was mixed with HisPur Ni-NTA Resin (Cytiva) and the target protein was combined with Ni Resin. Then the beads wash the beads 3 times with washing buffer. The target protein was eluted and collected with eluent. Elution proteins were frozen in liquid nitrogen and then stored at -80 °C or used for acylation in vitro.

## Extraction of histone

Histones were extracted using a standard extraction protocol[44]. Briefly, cells were collected and rinsed with cold-cooled PBS containing 5 mM sodium butyrate twice (300 g for 10 min). After centrifugation, the supernatant was discarded under vacuum. Then, dissolve cells with 1 ml TEB extract solution, transfer the solution into 1.5 ml centrifuge tube, and place on ice for 10 min. After centrifugation, centrifuge and discard the supernatant. Wash HeLa cells with half of the volume of TBE extract, centrifuge, and discard the supernatant. Add 0.2 mol/L $H_2SO_4$ solution of about 500 μl to dissolve the protein and up-down at 4 °C on the turning apparatus. Centrifuge 16000 g at 4 °C for 10 min, and transfer the supernatant to the new 1.5 ml tube. Add the same volume of 50% TCA solution with protein solution to the tube. Take the sample on the ice for 40 min to precipitate completely Centrifuge the sample 16000 g at 4 °C. After discarding the supernatant, washing histone three times with 100% acetone. Precipitation at room temperature dry completely. Finally, store the sample at −20 °C.

## Immunoaffinity purification of HBO1 and complexes from HEK293T cells

The plasmid Flag-HBO1 or its scaffold protein Flag-JADE1, Flag-BRPF2 was transfected or co-transfected in HEK293T cells. After 48 h of transfection, cells were collected and the required HBO1 protein or its complex was extracted by immunoprecipitation (IP). Briefly, lysis buffer (50 mM Tris-HCl pH 7.5, 150 mM NaCl, 1% Triton X-100, 1 mM EDTA, 1 mM DTT, 8% glycerol, Protease inhibitors) lysed the cells in ice for 30 min. After centrifugation, the supernatant was incubated overnight with pre-washed M2 beads rotating at 4 °C. The beads were washed 3 times with lysis buffer, the elution buffer (20 mM Tris-HCl pH 7.5, 150 mM NaCl, 0.1% NP40, 1 mM DTT, 10% glycerin, protease inhibitor, and Flag peptide) eluted the bound proteins. Elution proteins were frozen in liquid nitrogen and then stored at −80 °C or used for acylation in vitro.

## In vitro histone lactylation and acetylation assay

In vitro histone lactylation and acetylation assay were performed as described[20]. Histones of HEK293T were extracted as substrates according to the protocol of extraction histone[44]. Recombinant histone H3 and H4 purchased in Biolabs (M2503S, M2504S). Reaction mixture (including 100 μM lactyl-CoA or acetyl-CoA, 2 μg histone, 2 μg enzyme or its complex) was incubated in a reaction buffer (25 mM Tris-HCl pH 8.0, 150 mM NaCl, 10% glycerol, 1 mM sodium butyrate, 1 mM DTT) at 37 °C for 1 h. The reaction was terminated with 2 × SDS loading buffer and the acylation of histone was detected by western blot.

## SILAC labeling and sample preparation

WT HeLa and HBO1-KO HeLa cells were cultured in DMEMs containing L-Arginine and L-lysine (100 mg/ml) (HBO1-KO) or L-Arginine and 13C⁶-lysine (WT), respectively. All cells were cultured for more than 7 generations, and the labeling efficiency was more than 98%. WT and HBO1-KO cells were collected and the total proteins of the cells were extracted. Equal amounts of proteins are mixed together and precipitated using trichloroacetic acid, the precipitated proteins were dissolved by 100 mM $NH_4HCO_3$ and digested overnight by trypsin (1:50) at 37 °C. The digestive products were added to the final concentration of 5 mM DTT and incubated for 1 h at 56 °C, 15 mM iodoacetamide was incubated in the dark at room temperature for 45 min, and then 30 mM cysteine was incubated at room temperature for 30 min to block excess iodoacetamide. Then secondary digestion with trypsin (1: 100) at 37 °C for 4 h. The product was desalted by SepPak C18 cartridges (Waters) and subsequently drained by the SpeedVac system (Thermo Fisher Scientific). Immunoaffinity enrichment of lysine lactylation was then performed.

## Immunoaffinity enrichment

As previously mentioned, the samples were desalted and drained for immunoaffinity enrichment. Briefly, the drained sample was dissolved by NETN buffer (50 mM Tris-HCl pH 8.0, 100 mM NaCl, 1 mM EDTA, 0.5% Nonidet P-40). The samples were incubated with pre-washed anti-L-lactyllysine antibody-conjugated protein A agarose beads (PTM Biolabs) overnight at 4 °C. The incubated beads were washed 3 times with NETN buffer, ETN buffer (50 mM Tris-HCl pH 8.0, 100 mM NaCl, and 1 mM EDTA) twice, and HPLC grade-$H_2O$ once. The enriched peptides were eluted 3 times with 1% trifluoroacetic acid. The eluted solution was drained and desalted by C18 ZipTips (Millipore Corp), then nano-HPLC-MS /MS analysis was performed.

## HPLC-MS/MS analysis for Kla

Samples were reconstituted in 0.1% formic acid and then injected into a nano-LC system (EASY-nLC™ 1200, Thermo Fisher, San Jose) using trap-elute mode. Solvent A was 0.1% formic acid in water, while solvent B was 0.1% formic acid in 80% acetonitrile. After loading into the trap column (Thermo Scientific Acclaim PepMap 100 C18, 75 μm-i.d., 2 cm-

long, 3 μm, 100 Å), all of the peptides were further separated by a home-packed 75 μm-i.d., 25 cm-long C18 (1.9 μm, Dr. Maisch GmbH, Ammerbuch, Germany) column at flow rate of 300 nL/min with different gradient settings based on sample types. An Orbitrap Eclipse Tribrid mass spectrometer supplied with a FAIMS Pro Interface (Thermo Fisher, San Jose) was employed for MS analysis. Spray voltage was set to 2.0 kV and ion transfer tube temperature at 320 °C. Combination of different FAIMS CVs (compensation voltage) were set to run data-dependent acquisition (DDA) mode of the most intense precursors for 1 s cycle to build a big cycle of 2 s or 3 s. The detailed parameters were shown in the supplementary Data 4. In general, the proteome samples were separated with longer gradient, and fragment ions were acquired using ion trap for deeper profiling.

### Database search and data filter criteria for Kla
The database search and filter criteria were performed according to reported research. Briefly, Uniprot human proteome database was searched by MaxQuant (v.1.5.5.1), and an overall false discovery rate for peptides of less than 1%. Peptide sequences were searched using trypsin specificity and by allowing a maximum of two missed cleavages. The minimal peptide length was set to seven. Carbamidomethylation on Cys was specified as fixed modification. Lactylation on lysine as well as oxidation of methionine and acetylation on the peptide N terminus were fixed as variable modifications. Mass tolerances were set at ±10 ppm for precursor ions and ±0.02 Da for MS/MS. Lactylated peptides with a score of <40 and localization probability of <0.75 were further excluded. After quantifying the protein expression levels of the samples, we normalized the ratios of all quantitative Kla peptides with the ratios of corresponding protein levels.

### CRISPR-Cas9 knockout and shRNA knockdown
Target sgRNA was cloned into lenti-CRISPR v2 vector and then co-transfected into HEK293T cells with psPAX2 and pMD2G plasmids to produce the corresponding virus. After infecting the virus to the desired target cells for 36 h, 2 μg/ml purinomycin drug screening cells were used until no cells died due to purinomycin, that is, knockout mixed clonal cell lines were obtained.

For shRNA knockdown, the shRNA of the target gene was cloned into the pLKO.1-PURO vector, and other practices were consistent with CRISPR-Cas9 knockout.

All sequences of sgRNA and shRNA involved in this paper can be obtained in Supplementary Data 5.

### CUT&Tag
CUT&Tag is operated according to the manufacturer's instructions for the Hyperactive Universal CUT&Tag Assay Kit for Illumina (TD903, Vazyme Biotech). In brief, WT and HBO1-KO HeLa cells were collected and counted for incubation with pre-treated ConA beads. Subsequently, cells are resuspended in an Antibody Buffer and incubated overnight at 4 °C with the corresponding primary antibody of HBO1 or H3K9la. The secondary antibodies were diluted in appropriate proportions and incubate them with the cells at room temperature. Subsequently, pA/G-Tnp transposons were rotated and incubated with the samples for 1 h to activate the translocase fragment DNA and extract DNA. DNA library was constructed with TruePrEP Index Kit V2 for Illumina (TD202, Vazyme Biotech). The library was purified by VAHTS DNA Clean Beads (N411, Vazyme Biotech) and sequenced by Illumina novaseq 150PE.

### RT-qPCR and pG-MNase CUT&RUN for PCR/qPCR
For RT-qPCR, the total RNA of cells was extracted by Trizol method, followed by cDNA synthesis of 1 μg of RNA using StarScript III All-in-one RT Mix with gDNA Remover Kit (GenStar). SYBR-Green-based qPCR was performed to measure the expression of genes. The mRNA expression level was normalized by the expression level of GAPDH.

For pG-MNase CUT&RUN for PCR/qPCR, follow the manufacturer's instructions for Hyperactive pG-MNase CUT&RUN Assay Kit for PCR /qPCR (HD101, Vazyme Biotech). In brief, the cells were collected and counted, and incubated with pre-treated ConA Beads Pro. Cells are resuspended in Antibody Buffer and incubated overnight at 4 °C with the primary antibody. The pG-MNase Enzyme was rotated at 4 °C and incubated for 1 h. After fragmentation, DNA was terminated and released. FastPure gDNA Mini Columns were used to extract DNA, and then quantitative detection by qPCR was performed. The expression levels of the genes were normalized with the Spike in levels that had been added.

The sequences of all qPCR primers mentioned in this paper can be obtained in the supplementary data 5.

### CUT&Tag data processing and normalization
Bioinformatics analysis of CUT&Tag data: Sequencing quality was evaluated by FastQC version (v0.11.9). All reads were aligned to the human genome build hg38 using the Bowtie2 (v2.4.5) with the default parameters. Duplicate reads were removed using Picard tools (version 2.27.4). Then Samtools version (1.15.1) was used to convert and sort files to bam format. Uniquely mapped reads were normalized using deepTools (version 3.5.1) with the parameters --normalizeUsing RPKM --binSize 50 to visualize CUT&Tag signals at specific genomic loci by IGV (version 2.15.1). Peaks were identified for each sample and biological replicate using MACS2 (version 2.2.7.1) with command line options "macs2 callpeak -q 0.05 -g hs -f BAM --nomodel". To examine the reproducibility between replicates and across conditions, the genome is split into 10-kb bins, and a Pearson correlation of the log2-transformed values of read counts in each bin was calculated from the R package. CUT&Tag heatmaps were generated with deepTools to show normalized read counts at the peak center ±3 kb.

### Statistics and reproducibility
All data involving statistics are presented as mean ± standard deviation (SD). Statistical analysis was performed using GraphPad Prism software (version 8.0). The statistical significance was analyzed by Student's $t$-test. The exact $p$-value is provided in the corresponding figure. $p < 0.05(*)$, $p < 0.01$ (**), $p < 0.001$ (***) and $p < 0.0001$ (****) indicate statistically significant changes. For data presented without statistics, the experiment was repeated at least three times to ensure reproducibility, unless otherwise stated.

### Reporting summary
Further information on research design is available in the Nature Portfolio Reporting Summary linked to this article.

## Data availability
Source data are provided with this paper. The mass spectrometry proteomics data have been deposited to the ProteomeXchange Consortium via the PRIDE partner repository with the dataset identifier PXD051415. The CUT&Tag data generated in this study have been deposited in the Gene Expression Omnibus (GEO) repository under accession code GSE239656. The raw data generated in this study are provided in the Source Data file. All data are present in the paper, the Supplementary information, and Source Data file. Source data are provided with this paper.

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

## Acknowledgements

This work was supported by the Funding of National Natural Science Foundation of China to K.Z. (22074103, 22274114), C.L. (32200735), G.Z. (22374106), and X.B. (22004091) and the Talent Excellence Program from Tianjin Medical University to K.Z.

## Author contributions

K.Z. supervised experiments, Z.N., C.C., S.W. and K.Z. designed experiments. C.C., Z.N., C.L. and K.Z. wrote the manuscript. Z.N., S.W. and C.C. carried out cell culture, enzymatic activity assay, and molecular biological experiments. Z.N., S.W., C.L., J.Z., Y.H., Y. Z., and K.Z. carried out proteomic survey. Z.N., C.C., S.W. C.L., Z.W., A.W., J.M., Y.Y., Y. Z., C.H., X.B., S.T., G.Z., X.W. and K.Z. carried out

data collection, analysis, and interpretation. All of the authors discussed the results and commented on the manuscript.

## Competing interests

The authors declare no competing interests.
