## [Peer Review File · Nature Communications]

HBO1 catalyzes lysine lactylation and mediates histone H3K9la to regulate gene transcriptionEditorial Note: Parts of this Peer Review File have been redacted as indicated to remove third-party material where no permission to publish could be obtained.

REVIEWER COMMENTS

Reviewer #2 (Remarks to the Author):

Niu et al. hypothesized that MYST family KATs are lysin lactyltransferases and tested the hypothesis by using antibodies specific to lactylated lysines. They obtained a line of evidence that suggests HBO1 mediates lysine lactylation in cellulo and in vitro. Among the lactylated proteins, histone H3K9 stood out as a specific target of HBO1-mediated lactylation. H3K9 lactylation was decreased by knockout of HBO1 in Hela cells, which was coincided with a decrease of gene expression.

They claim to have made interesting discoveries;

(1)HBO1 lactylates a variety of proteins, especially H3K9.

(2)Lactylation of H3K9 may be implicated in gene regulation.

If these notions are true, I think this study is very valuable to the scientific community.

However, the study in the current form is not adequately convincing.

My questions and concerns are as follows.

Major questions

(1)Does HBO1 really directly lactylate histones?

The authors used anti-pan K1a and Kac antibodies in the first part of the paper, and later used specific antibodies to lactylated histones.

To confirm their observations and solidify their claims, the authors need to perform WB analysis using specific anti-lactylated histone H3K9 and 14, in every experiment they did with anti-pan K1a.

Especially, the in vitro lactylation assay data in Figure 2a needs to be verified with anti-H3K91a and H3K141a antibodies.

On the same note, acetylation of histones was shown using anti-pan Kac antibody, which gives a band pattern that does not corresponds to histones.

The mobilities do not seem to correspond with Histone H3 and H4.

I need to see the WB blots using anti-H3K14ac in every experiment where they used anti-pan Kac antibody.

(2)How abundant are lactylated histones compared to acetylated histones?

I wonder how much of histones are subjected to lactylation compared to acetylation.

Using MS data, the authors can give a rough estimate?

(3)The biological significance of histone lactylation by HBO1 remains unclear.

The authors discussed the biological significance of lactylation using the data of HBO1 KO cells.

However, HBO1 KO cause loss of H3K14 ac and other histone acetylation, which may be the reason for the observed reduction of gene expression.

We don't know if it was caused by the lack of lactylation.

I don't think they can say that HBO1 mediates a histone K1a-dependent gene transcription, as stated in the title and abstract.

Can the authors make lactyl Co-A-deficient cells, which may help dissecting the role of lactylation?

Minor points,

Figure 2b is particularly not convincing.

The authors should use recombinant histones that have no modifications as the substrate.

In Figure 2e and F, what molecular ratio does the x-axis represent?

HBO1 is known to acetylate Histone H3K14.

Why was it not identified as an endogenous K1a substrate of HBO1 in Figure 3d?

The description of in vitro histone lactylation assay is unclear.
How were the substrate histones prepared?

Figure 4 should contain H3K14ac blots as a positive control.

The authors have HBO1 KO cells and HBO1 expression vectors for WT and E508Q mutant.
A rescue experiment in Figure 4 would strengthen the paper.

There are some typos/grammatical errors in the manuscript that need to be corrected.

Line 34, cases,

Line 67, server

Line 83, Accumulation

Line 107, showed

Line 214, In consistent with

Line 292, unsure meaning....

Reviewer #3 (Remarks to the Author):

The authors elucidated the role of HBO-1 as a writer of lysine lactylation on histones, selected H3K9 as a representative substrate through a global lactylome study and presented a cellular phenotype. Since lactylation was proposed as a possible new epigenetic regulator on histones in 2019, this study is considered novel and valuable for its identification of the actual epigenetic role of lactylation. The research strategy of the study to elucidate the epigenetic role of lactylation is clear and logically organized. I have a few comments on the paper.

When performing a global lactylome analysis between HBO1-KO and WT, when selecting potential substrates of HBO1, how many lactylated peptides were detected that were only detected in WT? Were they included in the substrates of HBO1? The reduced ratio of HBO1/WT was determined to be a substrate of HBO1. Similarly, those detected only in WT are likely to be strong substrates of HBO1.

In the ms/ms spectra shown in Fig. 3 and Fig. S6, it is also necessary to address (1) is K on sequences a misscleavage? (2) please provide an ms/ms spectrum of an unmodified peptide corresponding to the sequence shown, or an ms/ms of a differently modified peptide at the same position. (3) If the analysis was performed by HR-mass, the m/z value should be shown to 3 decimal places.

Line 315, I think it would be better to limit the quoted part to protein acylation on histones.

This study showed reduced migration at Hela by HBO1-induced H3K9la. However, I believe that this is a phenotype demonstrated in an artificial KO cell model. I wonder if migration/invasion in any other cancer cells with different metastatic (aggressive) behavior correlates with increased H3K9la in response to increased HBO1? In clinical tissues, are there any differences in the expression of HBO1 and H3K9la between different stages of progression?

Reviewer #2 (Remarks to the Author):

Niu et al. hypothesized that MYST family KATs are lysin lactyltransferases and tested the hypothesis by using antibodies specific to lactylated lysines. They obtained a line of evidence that suggests HBO1 mediates lysine lactylation in cellulo and in vitro. Among the lactylated proteins, histone H3K9 stood out as a specific target of HBO1-mediated lactylation. H3K9 lactylation was decreased by knockout of HBO1 in Hela cells, which was coincided with a decrease of gene expression.

They claim to have made interesting discoveries;

(1) HBO1 lactylates a variety of proteins, especially H3K9.

(2) Lactylation of H3K9 may be implicated in gene regulation.

If these notions are true, I think this study is very valuable to the scientific community. However, the study in the current form is not adequately convincing. My questions and concerns are as follows.

Response: We are highly thankful to the reviewer for carefully evaluation and positive comments to our manuscript. We have addressed all the issues raised by the reviewer in the following point-to-point responses, added accordingly experiments and discussion in the revised manuscript.

Major questions

Comment 1: Does HBO1 really directly lactylate histones?

The authors used anti-pan K1a and Kac antibodies in the first part of the paper, and later used specific antibodies to lactylated histones. To confirm their observations and solidify their claims, the authors need to perform WB analysis using specific anti-lactylated histone H3K9 and 14, in every experiment they did with anti-pan K1a. Especially, the in vitro lactylation assay data in Figure 2a needs to be verified with anti-H3K91a and H3K141a antibodies.

On the same note, acetylation of histones was shown using anti-pan Kac antibody, which gives a band pattern that does not corresponds to histones. The mobilities do not seem to correspond with Histone H3 and H4. I need to see the WB blots using anti-H3K14ac in every experiment where they used anti-pan Kac antibody.

Response: We thank the reviewer for the constructive comments and suggestions. In this work, we used a quantitative proteomics strategy to identify potential HBO1 substrate proteins and found that H3K91a was significantly regulated by HBO1 (H3K141a was not characterized, possibly owing to its low abundance, also please see response to comment 6). Therefore, we focused on the regulation of H3K91a by HBO1. As suggested, we performed WB analysis using the histone H3K91a antibody in every experiment we did with anti-pan K1a to confirm our results, including the in vitro lactylation assay. We added the supplementary H3K91a results in (Fig. 4, Fig. 5, supplement fig. 7a-c,) to further verify that HO1 preferred to catalyze H3K91a.

We adjusted Figure 2b and labeled the positions of histone H3, H2A/2B, and H4. Following the reviewer's suggestion, we also added H3K14ac as a positive control in each result involving anti-Pan-Kac in the paper, further supporting the results.

Comment 2: How abundant are lactylated histones compared to acetylated histones? I wonder how much of histones are subjected to lactylation compared to acetylation. Using MS data, the authors can give a rough estimate?

Response: This is a good question. To our knowledge, there is currently no clear and accurate information on the abundance of lactylated histones compared to acetylated histones. To compare the abundance of lysine lactation and lysine acetylation on histones, we designed an experiment to quantify their total abundance on histones, referring to the previous report (Anal Chem, 2017, 17; 89 (2): 1299-1306). We first extracted histone proteins from HeLa cells and then enzymolized the histones into single amino acids using trypsin and Aminopeptidases. Next, we used HPLC-MS/MS for targeted identification of lactylated lysine and acetylated lysine. Finally, we obtained the abundance of lactylated lysine and acetylated lysine on histones. As shown in following Figure.1, the abundance of lactylated lysine in histones is about 2×10^5 , while the abundance of acetylated lysine is about 1×10^8 (about 500 times). The characteristic fragmentation ions of lactylated and acetylated lysine can confirm their attributes (Figure 1). We also noticed that the ratio of abundance of acetyl-CoA to lactyl-CoA is about 1000 times in the previous report (Varner, et. al. Open Biol. 2020, 10, 200187, as shown in the following table). So, we believed our estimated result is reasonable. Considering that this experiment is more relevant to our other work on analytical methods, we would like to report it in future work.

Figure.1 The intensity and ms/ms of lactylated lysine (a) and acetylated lysine (b) in histone proteins

[redacted]

(From Varner, et. al. Open Biol. 2020, 10, 200187)

Comment 3: The biological significance of histone lactylation by HBO1 remains unclear. The authors discussed the biological significance of lactylation using the data of HBO1 KO cells.

However, HBO1 KO cause loss of H3K14ac and other histone acetylation, which may be the reason for the observed reduction of gene expression. We don't know if it was caused by the lack of lactylation. I don't think they can say that HBO1 mediates a histone K14-dependent gene transcription, as stated in the title and abstract. Can the authors make lactyl Co-A-deficient cells, which may help dissecting the role of lactylation?

Response: We thank the reviewer for pointing out the question. Since HBO1 also regulates histone acetylation, there is a possibility that the relevant changes after HBO1-KO are not K14-dependent. To further study K14-dependent gene transcription, we performed two independent experiments. Lactic acid (LA) has been considered as the precursor source of lactyl-CoA, which could be transferred to substrates by lactyltransferase. We used oxamate to inhibit LDHA activity, which could lead to decrease of LA concentration in cells, as well as added sodium lactate externally to increase LA concentration in cells. Our results showed that in both conditions, histone lactylation (including H3K91a) but not acetylation was affected (Supplementary Fig. 10a and d). We further studied the gene changes regulated by H3K91a by CHIP-QPCR and verified that the genes *AQP1*, *LAMC2*, and *F10* discussed in our work could indeed be regulated by H3K91a. We added the experiments (Supplementary Fig. 10b, c, e, f) and discussion in the revised manuscript. We appreciate the reviewer's suggestion, which strengthens our work.

Minor points,

Comment 4: Figure 2b is particularly not convincing.

The authors should use recombinant histones that have no modifications as the substrate.

Response: Thank you very much for your professional question. Since the original figure

2b showed a shallow band of H4, the position of H2A/2B band was easily regarded as H4. We have adjusted the result and marked the position of H3, H2A/2B, and H4 band in the figure.

In addition, we performed in vitro enzyme activity verification of HBO1 with unmodified histones H3 and H4, excluding the influence of background modification of histones in acid-extracted cells, and found that HBO1, but not its mutant E508Q, had catalytic activity of histone lactation, consistent with our previous results (Fig. 2c and d). We added the above experiment and description in the revised manuscript.

Comment 5: In Figure 2e and F, what molecular ratio does the x-axis represent?

Response: The X-axis in this figure represents the molar ratio of the titrated small molecule to the protein being titrated. The molar ratio between small molecules and proteins in the ITC experiment gradually increases with the injection of small molecules, and this molar ratio is represented on the X-axis.

Comment 6: HBO1 is known to acetylate Histone H3K14. Why was it not identified as an endogenous K1a substrate of HBO1 in Figure 3d?

Response: Thanks for this question. We checked the published K-1a datasets, and H3K141a is not widely identified by LC-MS/MS. In the first reported histone lactylation work (Zhang D, et al. Nature, 2019, 575), lactylation was not identified on histone H3K14 in HeLa cells. In our data, H3K141a was also not identified in HeLa cells, which might be due to the low abundance of H3K141a.

Comment 7: The description of in vitro histone lactylation assay is unclear. How were the substrate histones prepared?

Response: In brief, the methods used in the in vitro histone lactylation assay and the preparation of histone proteins were carried out according to the published protocol (1. Xiao Y, et al. Nucleic Acids Res, 2021, 8037; 2. Liu X, et al. Cell Discov, 2017, 17016). The details of in vitro histone lactylation assay and extraction of histone has been added to the method. In addition, we used the commercial recombinant H3 and H4 without any modifications as the substrates to perform in vitro experiments of histone lactylation and acetylation (as suggested in comment 4). The experiments and corresponding results are added to the revised manuscript (Fig. 2c and d).

Comment 8: Figure 4 should contain H3K14ac blots as a positive control.

Response: Following the reviewer's suggestion, we added the positive control in Figure 4. In addition, H3K14ac was also added as a positive control in the results involving pan-Kac in the revised manuscript.

Comment 9: The authors have HBO1 KO cells and HBO1 expression vectors for WT and E508Q mutant. A rescue experiment in Figure 4 would strengthen the paper.

Response: We thank the reviewer's constructive criticism. We used the obtained HBO1-KO cell to carry out rescue experiments of HBO1-WT and E508Q and found that, consistent with our previous results, overexpression of HBO1-WT could restore the histone

K1a to the original modification level, while overexpression of HBO1-E508Q would lose the activity. The rescue experiment has been added to Figure 4. It further demonstrated that HBO1 has catalytic activity in regulating histone lactylation.

Comment 10: There are some typos/grammatical errors in the manuscript that need to be corrected.

Line 34, cases,

Line 67, server

Line 83, Accumulation

Line 107, showed

Line 214, In consistent with

Line 292, unsure meaning....

Response: Thank you very much for pointing out the typos and grammatical errors in the manuscript. We have corrected them in the revised manuscript and double-checked our manuscript.

Reviewer #3:

The authors elucidated the role of HBO-1 as a writer of lysine lactylation on histones, selected H3K9 as a representative substrate through a global lactylome study and presented a cellular phenotype. Since lactylation was proposed as a possible new epigenetic regulator on histones in 2019, this study is considered novel and valuable for its identification of the actual epigenetic role of lactylation. The research strategy of the study to elucidate the epigenetic role of lactylation is clear and logically organized. I have a few comments on the paper.

Response: We appreciate the reviewer for the positive appraisal of our work. We have addressed all the issues raised by the reviewer in the following point-to-point responses, added accordingly experiments and discussion in the revised manuscript.

Comment 1: When performing a global lactylome analysis between HBO1-KO and WT, when selecting potential substrates of HBO1, how many lactylated peptides were detected that were only detected in WT? Were they included in the substrates of HBO1? The reduced ratio of HBO1/WT was determined to be a substrate of HBO1. Similarly, those detected only in WT are likely to be strong substrates of HBO1.

Response: We thank the reviewer for the constructive comments. Following the reviewer's suggestions, we reanalyzed our dataset retrieved by MaxQuant. However, we did not find the K1a peptides detected only in WT. This situation may be caused by the limitation of MaxQuant's quantitative analysis strategy for SILAC data (Huang X, J Proteome Res. 2011 Mar 4; 10(3):1228-37.), which can lead to the loss of some peptides, especially those with low abundance. In the meanwhile, the quantification results for SILAC obtained by MaxQuant are considered to be more accurate with a high signal-to-noise ratio. Therefore, we preferred to use MaxQuant to analyze the SILAC data, and HBO1-regulated lactylated modified substrates were identified based on the abundance changes of K1a peptides.

In order to find potential substrates regulated by HBO1, we searched SILAC data separately with light lysine label and heavy lysine (6C13) label, respectively, and compared the two results. We found 54 lactylated peptides in WT but not HBO1-KO, as shown in the following table. However, considering the rigor of quantitative analysis and the consistency of MS data analysis, we still hope to use SILAC's quantitative data as the basis for subsequent analysis.

Kla peptide identified in HeLa cells but not in HBO1-KO cells.

Proteins	Positions within proteins	Protein names	Gene names	PEP	Score	Lactylation Probabilities
A8K8P3	26	Protein SF11 homolog	SF11	0.040575	45.718	MEK(1)K(1)VDSR
A8K8P3	27	Protein SF11 homolog	SF11	0.040575	45.718	MEK(1)K(1)VDSR
O15446	70	DNA-directed RNA polymerase I subunit RPA34	CD3EAP	0.01468	64.64	HVPLSGSQIVK(0.848)GK(0.152)
O60229	529	Kalirin	KALRN	0.078649	41.029	LESIWQHRK(1)
O60684	18	Importin subunit alpha-7	KPNA6	0.019018	62.528	SYK(1)NINALPEEMR
P62807;P57053;O60814	17;17;17;17;17;17	Histone H2B type 1-C/E/F/G/I;Histone H2B type F-S;Histone H2B type 1-K;Histone H2B type 1-L;Histone H2B type 1-O;Histone H2B type 1-D;Histone H2B type 1-H;Histone H2B type 1-N	HIST1H2BC;H2BFS;HIST1H2BK;HIST1H2BL;HIST1H2BO;HIST1H2BD;HIST1H2BH;HIST1H2BN	0.057211	74.486	K(1)AVTK(1)AQK
Q99880;P23527;P58876;Q93079;Q90877	21;21;21;21;21;21	Histone H2B type 1-C/E/F/G/I;Histone H2B type F-S;Histone H2B type 1-K;Histone H2B type 1-L;Histone H2B type 1-O;Histone H2B type 1-D;Histone H2B type 1-H;Histone H2B type 1-N	HIST1H2BC;H2BFS;HIST1H2BK;HIST1H2BL;HIST1H2BO;HIST1H2BD;HIST1H2BH;HIST1H2BN	0.057211	74.486	K(1)AVTK(1)AQK
P62807;P57053;O60814	6;6;6	Histone H2B type 1-C/E/F/G/I;Histone H2B type F-S;Histone H2B type 1-K	HIST1H2BC;H2BFS;HIST1H2BK	0.027182	63.944	PEPAK(1)SAPAPK
O75122	709	CLIP-associating protein 2	CLASP2	0.06627	46.37	MFADPHGK(1)
O76021	374	Ribosomal L1 domain-containing protein 1	RSL1D1	0.017575	48.677	ATNESEDEIPOLVPIGK(0.458)K(0.542)
P18206	381	Vinculin	VCL	0.078655	43	LEAMTNSK(1)
P19338	124	Nucleolin	NCL	0.061241	77.923	ALVATPGK(0.5)K(0.5)
P19338	125	Nucleolin	NCL	0.061241	77.923	ALVATPGK(0.5)K(0.5)
P22061	4	Protein-L-isoaspartate(D-aspartate) O-methyltransferase	PCMT1	0.006631	55.589	AWK(1)SGGASHSELHNLR
P22314	8	Ubiquitin-like modifier-activating enzyme 1	UBA1	0.031782	48.283	SSSPLSK(0.5)K(0.5)
P23527;Q93079;Q5QN	6;6;6	Histone H2B type 1-O;Histone H2B type 1-H;Histone H2B type 2-F	HIST1H2BO;HIST1H2BH;HIST1H2BF	0.014606	78.043	PDPAK(1)SAPAPK
P28838	455	Cytosol aminopeptidase	LAP3	1.23E-05	103.77	QVVDLQDLADVNNIGK(1)YR
P33981	192	Dual specificity protein kinase TTK	TTK	0.040625	42.743	K(1)QLLSFEEKK
P46777	276	60S ribosomal protein L5	RPL5	0.016894	138.48	MILAQK(0.902)K(0.098)
P68104;Q5VTE0	84;84	Elongation factor 1-alpha 1;Putative elongation factor 1-alpha-like 3	EEF1A1;EEF1A1P5	0.037491	49.482	GITDLSLWK(0.357)FETSK(0.643)
P78527	2908	DNA-dependent protein kinase catalytic subunit	PRKDC	0.050804	67.897	LLPAELPAK(1)R
Q09666	1208	Neuroblast differentiation-associated protein AHNAK	AHNAK	0.011046	65.092	ISMSPVDLHLLK(0.421)GPK(0.579)
Q09666	1333	Neuroblast differentiation-associated protein AHNAK	AHNAK	0.010205	59.898	ISMSPVDLNLK(0.864)GPK(0.136)
Q12789	530	General transcription factor 3C polypeptide 1 Serine/threonine-protein kinase PAK 2;PAK-2p27;PAK-2p34	GTF3C1	0.05135	47.603	VVNLHPLK(0.435)K(0.565)
Q13177	52		PAK2	0.006568	100.19	HK(1)IISIFSGTEK
Q13596	445	Sorting nexin-1	SNX1	0.076269	44.246	LLWANK(0.002)PDK(0.999)LQQAQ(0.999)
Q13596	450	Sorting nexin-1	SNX1	0.076269	44.246	LLWANK(0.002)PDK(0.999)LQQAQ(0.999)
Q13838	36	Spliceosome RNA helicase DDX39B	DDX39B	0.000343	86.641	DVK(1)GYSYVHSSGFR
Q14103	221	Heterogeneous nuclear ribonucleoprotein D0	HNRNPD	0.014258	58.964	EYFGGFGVEVESIELPMDNK(0.5)TNK(0.5)
Q15269	700	Periodic tryptophan protein 2 homolog	PWP2	0.0625	46.663	HFK(1)PEIR
Q16666	451	Gamma-interferon-inducible protein 16	IFI16	0.005768	81.278	SEDTSK(1)MNFDMR
Q2KHR3	1479	Glutamine and serine-rich protein 1	QSER1	0.037464	56.551	TTTTTK(1)APSVKPK
Q5JCQ9	706	A-kinase anchor protein 4	AKAP4	0.011868	49.929	LVESVMK(0.855)LCLIMAK(0.145)
Q62MW3	1082	Echinoderm microtubule-associated protein-like 6	EML6	0.023733	46.819	K(1)EMISDIK(0.076)FSK(0.924)
Q62MW3	1092	Echinoderm microtubule-associated protein-like 6	EML6	0.023733	46.819	K(1)EMISDIK(0.076)FSK(0.924)
Q7L014	271	Probable ATP-dependent RNA helicase DDX46	DDX46	0.066952	111.17	VVTVVTTK(0.5)K(0.5)
Q7L014	272	Probable ATP-dependent RNA helicase DDX46	DDX46	0.066952	111.17	VVTVVTTK(0.5)K(0.5)
Q9H8S9;Q7L9L4	17;17	MOB kinase activator 1A;MOB kinase activator 1B	MOB1A;MOB1B	0.000874	95.428	K(1)NIPESHQYELLK
Q8NC51	68	Plasminogen activator inhibitor 1 RNA-binding protein	SERBP1	0.013351	59.27	SAAQAAQTNSNAAGK(1)QLR
Q8NFM4	267	Adenylate cyclase type 4	ADCY4	0.003967	54.103	LQAGQSRPESTNNFHSLYVK(1)
Q92922	346	SWI/SNF complex subunit SMARCC1	SMARCC1	0.059258	113.18	K(1)GQASLYGK
Q96DE0	190	U8 snoRNA-decapping enzyme	NUDT16	2.41E-06	68.386	EQLLEALQDLGLLQSGSISGLK(1)IPAHHI
Q9BYT9	172	Anoctamin-3	ANO3	0.045958	42.109	K(1)TNIQYDK(1)
Q9BYT9	179	Anoctamin-3	ANO3	0.045958	42.109	K(1)TNIQYDK(1)
Q9C0G6	2873	Dynein heavy chain 6, axonemal	DNAH6	0.037676	42.109	VEK(1)VSK(1)ACK(1)
Q9C0G6	2876	Dynein heavy chain 6, axonemal	DNAH6	0.037676	42.109	VEK(1)VSK(1)ACK(1)
Q9C0G6	2879	Dynein heavy chain 6, axonemal	DNAH6	0.037676	42.109	VEK(1)VSK(1)ACK(1)
Q9H1E3	184	Nuclear ubiquitous casein and cyclin-dependent kinase substrate 1	NUCKS1	0.072351	113.71	ATVTPSPVK(0.748)GK(0.252)
Q9H267	324	Vacuolar protein sorting-associated protein 33B	VPS33B	0.069802	43.808	GMDIKQMK(1)
Q9NR45	355	Sialic acid synthase	NANS	0.055705	41.256	VLVTVEEDDTIMEELVDNHGK(0.5)K(0.5)
Q9NR45	356	Sialic acid synthase	NANS	0.055705	41.256	VLVTVEEDDTIMEELVDNHGK(0.5)K(0.5)
Q9Y2S6	59	Translation machinery-associated protein 7	TMA7	0.041341	103.56	GPLATGGIK(0.5)K(0.5)
Q9Y2S6	60	Translation machinery-associated protein 7	TMA7	0.041341	103.56	GPLATGGIK(0.5)K(0.5)
Q9Y3C8	122	Ubiquitin-fold modifier-conjugating enzyme 1	UFC1	0.013124	56.951	ICLTDHFK(1)PLWAR

Comment 2: In the ms/ms spectra shown in Fig. 3 and Fig. S6, it is also necessary to address (1) is K on sequences a misscleavage? (2) please provide an ms/ms spectrum of an unmodified peptide corresponding to the sequence shown, or an ms/ms of a differently modified peptide at the same position. (3) If the analysis was performed by HR-mass, the m/z value should be shown to 3 decimal places.

Response: Thanks for your valuable suggestions. (1) We have marked the missing cut sites of the peptides we have shown in the diagram. (2) We selected and added the ms/ms spectrum of the same peptides modified with lactation and acetylated peptides at the same site, which were listed in supplement fig. 6c, (3) and the m/z values were shown to three decimal places.

Comment 3: Line 315, I think it would be better to limit the quoted part to protein acylation on histones.

Response: Thanks for your suggestion. We have modified this part of the article to limit the acylation of histones.

Comment 4: This study showed reduced migration at Hela by HBO1-induced H3K9la. However, I believe that this is a phenotype demonstrated in an artificial KO cell model. I wonder if migration/invasion in any other cancer cells with different metastatic (aggressive) behavior correlates with increased H3K9la in response to increased HBO1? In clinical tissues, are there any differences in the expression of HBO1 and H3K9la between different stages of progression?

Response: We thank the reviewer for the constructive comments. Following the reviewer's suggestions, we added a series of experiments to demonstrate that HBO1-regulated H3K9la enhances cancer migration and invasion abilities.

(1) HBO1 knockout was performed on six other cancer cell lines in addition to HeLa, including HepG2 (hepatocellular carcinoma), U87MG (gliomas), KYSE-30 (esophageal squamous cell carcinoma), MDA-MB-231 (breast carcinoma), HCT116 (colon cancer), and H460 (non-small cell lung cancer). Similarly, HBO1 knockout leads to the decrease of histone H3K9la level. We also conducted a set of experiments, including clonal formation, wound healing, and transwell assay on these HBO1-knocked out cancer cell lines and found limited cell proliferation, migration and invasion abilities compared to wild-type cells. (2) Furthermore, we examined the levels of H3K9la and HBO1 expression in clinical samples of cervical cancer samples. 84 cervical cancer cell samples and 52 normal cervical cell samples were included in our study. As expected, both H3K9la and HBO1 expression in cancer cells were significantly higher than those in normal tissues. In addition, the level of H3K9la modification was positively correlated to the expression of HBO1 in cervical cancer samples. Therefore, H3K9la is associated with HBO1 and promotes the cervical tumorigenesis.

We have added the experiments (Fig. 6i, j, k, l and supplemental Fig 11) and modified abstract, method, results and discussion in the revised manuscript. We are grateful for the reviewer's suggestions, which improve the quality of our work.

REVIEWERS' COMMENTS

Reviewer #2 (Remarks to the Author):

The authors addressed my questions and concerns.
I have no further comments.

Reviewer #3 (Remarks to the Author):

The authors have revised the text in response to reviewer comments and conducted supplemental experiments. In particular, the involvement of H3K9la by HBO1 was clearly demonstrated, and furthermore, H3K9la-induced phenotypes were identified in cellular systems. We have no further comments on the revised paper and believe it is sufficiently good to be published in Nat Com.